# MuVi: Video-to-Music Generation with Semantic Alignment and Rhythmic Synchronization

## Abstract

Generating music that aligns with the visual content of a video has been a challenging task, as it requires a deep understanding of visual semantics and involves generating music whose melody, rhythm, and dynamics harmonize with the visual narratives. This paper presents MuVi, a novel framework that effectively addresses these challenges to enhance the cohesion and immersive experience of audio-visual content. MuVi analyzes video content through a specially designed visual adaptor to extract contextually and temporally relevant features. These features are used to generate music that not only matches the video's mood and theme but also its rhythm and pacing. We also introduce a contrastive music-visual pre-training scheme to ensure synchronization, based on the periodicity nature of music phrases. In addition, we demonstrate that our flow-matching-based music generator has in-context learning ability, allowing us to control the style and genre of the generated music. Experimental results show that MuVi demonstrates superior performance in both audio quality and temporal synchronization. The generated music video samples are available at muvi-v2m.github.io.

## 1 Introduction

The development of multimedia social platforms and AI-generated content (AIGC) in recent years has fundamentally changed the way people engage with video content. Music, an essential element of videos, has spurred considerable interest in video-to-music generation technology. The video-to-music (V2M) task focuses on generating matching music based on the visual content of video, offering immense potential in fields such as advertising and video content creation.

The generated music for the video should exhibit two key qualities: semantic alignment and rhythmic synchronization. Semantic alignment ensures that the generated music captures the emotional and thematic essence of the video content, while rhythmic synchronization ensures that the music's tempo and rhythm are in harmony with the visual dynamics. Achieving both is crucial for creating a cohesive audio-visual experience.

Previous V2M methods (Zhuo et al., 2023; Kang et al., 2024; Li et al., 2024c) primarily focus on generating music that aligns with the global features (theme, emotion, style, etc.) of the entire video clip. Other methods focus only on semantic alignment (Tian et al., 2024) or on rhythmic synchronization (Su et al., 2024). However, when video shifts from one theme or emotion into another, the generated music should adapt accordingly in both terms of semantic and rhythm. The most relevant research on generating synchronized music in accordance with video dynamics can be found in studies that endeavor to create music based on human movement within dance videos(Zhu et al., 2022; Era et al., 2023; Li et al., 2024b). However, the topic is limited to dancing, whose visual semantics are relatively simple to capture. Research on generating music that seamlessly adapts to the content and style of videos in general are still absent.

In this paper we aim to tackle these long-standing challenges of video-to-music generation:

- **Semantic alignment.** It is anticipated that the style, melody, and emotions of the music will evolve in harmony with the video content. Consider this scenario: Tom from *Tom and Jerry* (Hanna & Barbera, 1940) is peacefully sleeping under a tree when suddenly an apple falls on his head. Initially, the music may have a soothing tone, but within just a few seconds,

the mood of the music immediately becomes intense due to the sudden event of the apple falling, and the music instruments may switch to the ones with more intense timbre.

- **Rhythm synchronization.** In common, music typically has a relatively stable rhythm, which tends to remain constant in a musical section. However, in video soundtracks, the rhythm often changes instantaneously in response to the dynamics of video content, especially to the motion of the characters. The music's beat is often synchronized with the movements of characters. Consider the previous scenario, when Tom is sleeping, the music's beat should synchronize with the pace of snoring; and the moment the apple hits his head, a drumbeat occurs and the rhythm suddenly becomes clipped and harsh. This requires precise frame-level synchronization between video and music. Relying only on global understanding of the video may lead to audio-visual asynchrony.

It is also worth mentioning that, unlike regular music, video soundtracks often contain special sound effects mimicked by musical instruments. Still considering the previous scenario, the moment the apple hits Tom's head might be accompanied by a loud crash cymbal or snare drum to simulate the impact sound. Moreover, the objects appearing in the video might not be directly related to generated music or sound. For example, when Jerry hits Tom with a violin, it doesn't necessarily imply violin music. This renders some traditional video-sound or video-music datasets ineffective, as they are constructed based on the pairing of the sound-producing object and the sound produced.

To achieve precise music-visual synchronization, we need a high-frame-rate method for video semantic extraction that captures feature sequences with a small time span. Simultaneously, this video feature sequence should contain guiding information about the accurate positions of music beats. A natural idea is to leverage contrastive learning to model synchronization, like Diff-Foley (Luo et al., 2024). However, unlike contrastive audio-visual learning in Diff-Foley, music has a higher temporal density and a more complex spectrum. Constructing negative samples by mismatched music-video pairs might lead to inefficient learning, as the model could take shortcuts and overfit to more easily identified features.

This paper presents MuVi, a novel V2M method that generates music soundtracks for videos. The main model features a simple non-autoregressive encoder-decoder architecture, where the encoder is one of the open-source pre-trained visual encoders, and the decoder is adapted from a pre-trained flow-matching based music generator (Lipman et al., 2022). We propose a visual adaptor to connect the visual encoder and the music generator, which efficiently compresses high-frame-rate visual features to provide sufficient temporal resolution for accurate synchronization. For rhythmic synchronization, we apply contrastive pre-training on the adaptor at first to enhance its perception of rhythmic motion and penalize asynchronous video-music pairings. In addition to using mismatched music and video pairs, we employ two strategies to construct negative samples: 1) Temporal shifting, where we randomly shift the original music along the timeline to create asynchrony; 2) Random replacement, where we replace a randomly selected segment by other music segment or silence. It is worth mentioning that the pre-trained visual adaptor and audio encoder also serve as an evaluation metric for synchronization.

Experimental results show that the proposed method demonstrates superior performance over baselines in terms of semantic alignment and rhythmic synchronization. We investigate the impact of various visual compression strategies on the downstream music generation task. We also demonstrate the potential of in-context learning capability and scalability.

The main contributions of this work are:

- We propose a video-to-music framework that effectively generates music soundtracks with both semantic alignment and rhythmic synchronization — a challenge that has not been thoroughly explored before.

- We introduce a visual adaptor that efficiently compresses visual information and ensures length alignment for the non-autoregressive music generator.

- We design a contrastive music-visual pre-training scheme that emphasizes the periodic nature of beats, forcing the model to focus on recognizing rhythmic asynchrony.

## 2 BACKGROUND

**Video-to-music Generation** Existing V2M methods can be categorized into several classes. 1) Earlier methods (Koepke et al., 2020; Gan et al., 2020; Su et al., 2020a;b) mostly focus on reconstructing the sound of instruments for silent instrumental performance videos, which is essentially more of a video-to-audio task; 2) Some works (Yu et al., 2023; Zhu et al., 2022; Aggarwal & Parikh, 2021; Lee et al., 2019) focus on generating music to match human motions (such as dance videos). These methods are restricted to only a few kinds of body motions such as dancing and figure skating, which are originally designed to match the tempo and style of the chosen music. They do not generalize well to any human movements or any video; 3) Recent methods such as (Di et al., 2021; Zhuo et al., 2023; Kang et al., 2024; Li et al., 2024c) focus on understanding general videos and generating symbolic music, where CMT (Di et al., 2021) generates chords and notes with rule-based video-music relationships, and SymMV (Zhuo et al., 2023) pays attention to semantic-level correspondence. However, symbolic music generation often requires costly annotated music score or MIDI data and thus cannot be scaled efficiently; 4) More recent works try to directly generate music waveforms from video without the need for symbolic data – VidMuse (Tian et al., 2024) models the video-music correlation using a long-short-term visual module, and V2Meow (Su et al., 2024) utilizes an autoregressive Transformer (Vaswani, 2017) to generate acoustic music tokens. $M^2$UGen (Hussain et al., 2023) leverages multiple multi-modal encoders and the power of large language models (LLM). However, these methods primarily model global semantic and emotional features of the entire video clip and lack the ability to effectively capture the sudden change in themes and emotions. To the best of our knowledge, generating music video soundtracks with both semantic alignment and rhythmic synchronization has not yet been thoroughly studied.

**Video-to-audio Generation** Although we mainly focus on V2M, music generation and video-to-audio (V2A) methods share many similarities. One of the most important aspects that both V2M and V2A emphasize is the audio-visual synchronization. Some methods leverage GAN for audio generation, with specially designed hand-crafted visual and motion features to encode video content (Chen et al., 2020; Ghose & Prevost, 2022; Iashin & Rahtu, 2021). Im2Wav (Sheffer & Adi, 2023) adopts contrastively pre-trained CLIP (Radford et al., 2021) visual encoder to extract semantic content to condition the music generation, while Diff-foley (Luo et al., 2024) proposes a novel visual feature obtained through contrastive audio-visual pre-training. Frieren (Wang et al., 2024) utilizes an ODE-based generator and direct audio-visual fusion to improve sound quality and synchronization, while FoleyCraft (Zhang et al., 2024) introduces sound event annotations and temporal supervision to further enhance alignment. However, the current audio-visual contrastive learning based on a simple mismatch-construction mechanism may not be suitable for music modality, where the complexity, both in the frequency and time domains, far exceeds that of audios studied in V2A tasks. Additionally, manual annotation of sound events may not be feasible for music.

**Music Generation** Our work is essentially a conditional music generation method. There are plenty of methods (Dhariwal et al., 2020; Agostinelli et al., 2023; Schneider et al., 2023; Huang et al., 2023; Forsgren & Martiros, 2022; Maina, 2023; Copet et al., 2024; Chen et al., 2024; Lan et al., 2024; Li et al., 2024a) focusing on generating acoustic music soundtracks, conditionally or unconditionally. MusicLM (Agostinelli et al., 2023) and MusicGen (Copet et al., 2024) model the music representations using language models (LM) with an autoregressive process. Music diffusion models (Huang et al., 2023; Schneider et al., 2023; Maina, 2023) utilize denoising diffusion probabilistic models (Ho et al., 2020) or latent diffusion strategy (Rombach et al., 2022) to improve audio quality and diversity, while MelodyFlow (Lan et al., 2024) leverages the ODE-based flow-matching mechanism. Since our focus is not on the music generator, we adopt a prevailing flow-matching-based architecture (Lipman et al., 2022), which is used in many mainstream music generation frameworks.

## 3 METHOD

This section introduces MuVi, whose main architecture is illustrated in Figure 1. MuVi consists of a visual encoder, an adaptor, and a music generator. Since the temporal length of the generated music has to be the same as the input video, a non-autoregressive music generator is a natural choice, as duration prediction is no longer necessary. To capture the sudden visual events for music-visual synchronization, a higher temporal resolution is required, although this is computationally demanding.

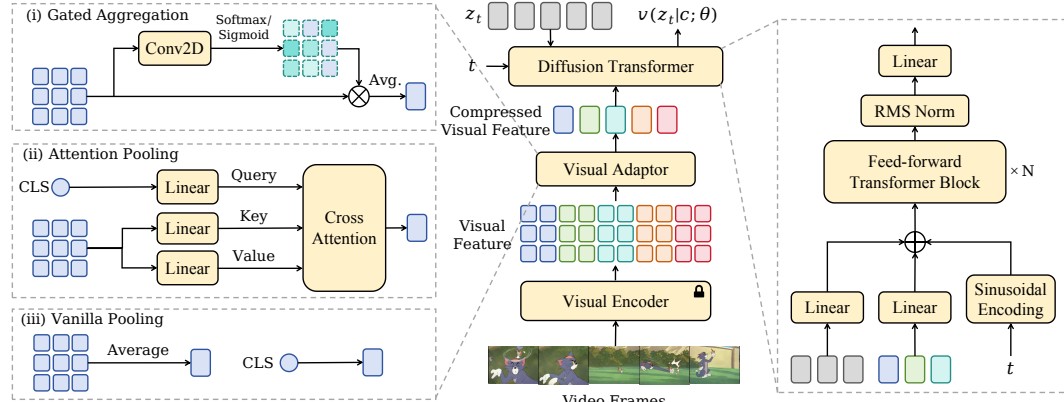

Figure 1: The architecture of MuVi. The main model and the input/output are illustrated in the middle, where the visual encoder is frozen during the training stage. The visual compression strategies are listed on the left, where "CLS" indicates the CLS token of certain visual encoders, such as CLIP. The architecture of the diffusion Transformer is illustrated on the right.

Furthermore, for each frame, multiple visual features are extracted, with each feature representing a different patch. To ensure feature length alignment in the non-autoregressive framework and increase efficiency, it is of our desire to compress the features, retaining visual information most related to the generated music. Therefore, we introduce a visual adaptor that efficiently compresses visual features while preserving enough information to achieve rhythmic synchronization and semantic alignment.

The section is organized as follows: we first discuss several aggregation strategies of visual representation, which is the key factor in the design of the visual adaptor. Finally, we introduce the contrastive pre-training strategy for the visual adaptor, followed by a description of the architecture of the music generator. Due to space constraints, the details of the training and inference pipelines are listed in Appendix B

## 3.1 VISUAL REPRESENTATION MODELING

CLIP (Radford et al., 2021), as the representative of the contrastive text-visual learning family, can generate image features rich in semantic space, and their visual transformer (ViT) (Alexey, 2020) encoders come with a CLS token to capture global information. VideoMAE (Tong et al., 2022) and VideoMAE V2 Wang et al. (2023) focus on self-supervised pre-training for a single video modality. As encoders, they are more unbiased compared to methods trained with semantic supervision and maintain an understanding of the temporal dimension. There are also audio-visual joint representations such as CAVP Luo et al. (2024), with lower frames per second (FPS).

Since we use a non-autoregressive generator, we need to compress the video features to align with the frame length of the audio features. And it is of our interest to ensure that the compressed feature carry as much visual and semantic information to guide music generation as possible. Given a patchified video representation sequence $x \in \mathbb{R}^{N \times L \times C}$, where $N$, $L$, $C$ are the number of frames, the number of patches (where $L = H \times W$), and the channel dimension, the resulting feature, $\bar{x} \in \mathbb{R}^{N \times C}$, should achieve a compression ratio of $L$ times, representing each frame by a single vector. We compare three major compression methods, which are illustrated on the left part of Figure 1.

**Gated Aggregation**  The gated aggregation method is essentially a dynamic weighting strategy. We use a shallow projection as the gating layer $g(\cdot)$ (typically a single-layer MLP or CNN) to transform $x$ into a global weight map for each frame: $w_i = \sigma(g(x_i))$, where $x_i$ is the $i$th frame of the video feature ($i \in \{1, ..., N\}$), $w_i \in \mathbb{R}^{L \times 1}$, and $\sigma(\cdot)$ is a non-linear function that ensures $w_i$ is between 0 and 1. If $\sigma(\cdot)$ is the Sigmoid function, then the aggregated feature can be computed as $\bar{x}_i = w_i^\top x_i / (\sum w_i)$. If $\sigma(\cdot)$ is Softmax, then $\bar{x}_i = w_i^\top x_i$. Gated aggregation is useful for ViTs without the CLS token, such as VideoMAE and VideoMAE V2, where the Softmax aggregation provides the capability to model global distributions.

**Attention Pooling**    For ViTs with the CLS token, a semantic aware attention pooling strategy can be established (Hou et al., 2022). Given a CLS token at the $i$the frame $c_i \in \mathbb{R}^C$ and the patches $x_i$, we compute the query $q_i \in \mathbb{R}^C$, the key $k_i \in \mathbb{R}^{L \times C}$, and the value $v_i \in \mathbb{R}^{L \times C}$ through three linear projections (Vaswani, 2017), where the query comes from the CLS token and key/value comes from the patches. Then we perform a single-query cross-attention to acquire the attention weights $w_i = \sigma(q_i k_i^\top / \sqrt{C})$, where $\sigma$ is Softmax function. The compressed feature is then $\bar{x}_i = w_i^\top x_i$. We believe that this strategy not only considers global features but also selectively captures local information.

**Vanilla Pooling**    Beyond the pooling strategies above, we also consider simpler pooling methods. For ViTs without the CLS token, a mean pooling strategy is computing the global average of each frame. For CLIP, we simply use the CLS token as the pooled representation. For audio-visual pre-trained representations, such as CAVP, we perform no operations (in fact, we need to upsample the CAVP representations temporally, to meet the length of music features).

## 3.2    Contrastive Music-visual Pre-training

The compressed visual features are instrumental for aligning the generated music to visual semantic information, such as the sequence of events and the event positions in the frame, but lack precise timestamps for exact rhythmic synchronization. To achieve more precise synchronization, we design a contrastive music-visual pre-training strategy.

To begin with, we adopt a contrastive loss approach to maximize the similarity of music-visual pairs from the same video while minimizing the similarity of pairs from different videos. To address temporality, we take the temporal position into consideration when computing the contrastive loss. Given a music-visual representation pair $(x_m, \bar{x}_v)$, where $x_m \in \mathbb{R}^{C' \times F \times N'}$ is acquired using a pre-trained audio encoder of AudioMAE (Huang et al., 2022), $F$ is the number of frequency bins, and $\bar{x}_v \in \mathbb{R}^{N \times C}$ is obtained using the visual adaptor. $x_m$ is averaged along the frequency axis, and then resampled along the temporal dimension to align with the video frames $N$, and finally transformed into $\bar{x}_m \in \mathbb{R}^{N \times C}$ with a learnable head. Therefore, the position-aware similarity function can be implemented as $\mathrm{SIM}(\bar{x}_m, \bar{x}_v) = \frac{1}{N} \sum_{t=1}^{N} \bar{x}_{m,t}^\top \bar{x}_{v,t} / (|\bar{x}_{m,t}| \cdot |\bar{x}_{v,t}|)$, based on the cosine similarity.

For a paired subset of the dataset, $\mathcal{B} = \{(\bar{x}_m^i, \bar{x}_v^i)\}_{i=1}^M$, where $M$ is the size of the subset and the pair within each tuple is considered positive, the contrastive objective is then adopted:

$$\mathcal{L}_S^{(i,j)} = -\frac{1}{2} \log \frac{\exp(\mathrm{SIM}(\bar{x}_m^i, \bar{x}_v^j)/\tau)}{\sum_{k=1}^M \exp(\mathrm{SIM}(\bar{x}_m^i, \bar{x}_v^k)/\tau)} - \frac{1}{2} \log \frac{\exp(\mathrm{SIM}(\bar{x}_m^i, \bar{x}_v^j)/\tau)}{\sum_{k=1}^M \exp(\mathrm{SIM}(\bar{x}_m^k, \bar{x}_v^j)/\tau)} \tag{1}$$

where $\mathcal{L}_S^{(i,j)}$ stands for the semantic objective, and the number of positive and negative pairs are $M$ and $M^2 - M$, respectively; each video/music feature in $\mathcal{B}$ corresponds to $M - 1$ negative music/video matches.

However, this loss may be insufficient because numerous factors, such as melody or musical instrument, can cause a music-visual pair to be considered a negative sample. This challenge makes it difficult for the model to focus on rhythmic synchronization, often causing it to overfit to other more easily identified features. Therefore, we introduce two new sets of negative pairs, forcing the model to focus on temporal synchronization:

- **Temporal shift.** For a positive pair $(\bar{x}_m^i, \bar{x}_v^i)$ in the subset $\mathcal{B}$, we randomly and temporally shift the original music waveform to obtain the shifted music feature $\tilde{x}_m^i$, thus creating asynchrony and constructing a new negative pair $(\tilde{x}_m^i, \bar{x}_v^i)$. It is worth mentioning that the shifting operation is not entirely random, because if a clip of music has a relatively stable tempo, randomly moving it by whole beats may not significantly affect the synchronization. Therefore, we leverage a dynamic beat tracking algorithm (Ellis, 2007) to obtain the minimum beats per minute (BPM) and its corresponding minimal cycle for this music clip. Specifically, if the minimal beat cycle is $n$ frames, we can only shift $kn + b$ in both directions, where $b \in \{\lceil 0.1n \rceil, .., \lfloor 0.4n \rfloor, \lceil 0.6n \rceil, ..., \lfloor 0.9n \rfloor\}$. $k$ is a random positive integer to make sure that $kn + m < 0.5N$, where $N$ is the total frame length. Note that we also intentionally skipped the half-beat areas to avoid backbeat synchronization. This approach initially creates $M$ negative pairs. To increase the number of negative samples, we

apply the same process to the originally mismatched negative pairs, ultimately obtaining $M^2$ negative pairs in total.

- **Random replacement.** Since we deconstruct the similarity measure temporally, the model needs to focus on every position on the time axis. If synchronization occurs only in most intervals and not all, the similarity should accordingly decrease. For a positive pair $(\bar{\boldsymbol{x}}_m^i, \bar{\boldsymbol{x}}_v^i)$ in $\mathcal{B}$, we construct another negative music sample, $\hat{\boldsymbol{x}}_m^i$, by replacing a random segment within the original waveform of $\bar{\boldsymbol{x}}_m^i$ with another music clip in $\mathcal{B}$, while retaining the parts outside the selected interval still. The segment length is randomly chosen from $0.2N$ to $0.4N$. When replacing music segments, we leave 5% of the length at both ends and use an equal-power crossfade transition for music waveforms belonging to different videos.

Therefore, each video feature corresponds to $2M$ additional negative matches, resulting in $3M^2 - M$ negative pairs. If it is impossible to construct negative samples using the above two strategies, then the negative music features are directly computed from silence. The objective is then modified as:

$$\mathcal{L}_T^{(i,j)} = -\frac{1}{2} \log \frac{\exp(\text{SIM}(\bar{\boldsymbol{x}}_m^i, \bar{\boldsymbol{x}}_v^j)/\tau)}{\sum_{k=1}^M \exp(\text{SIM}(\bar{\boldsymbol{x}}_m^i, \bar{\boldsymbol{x}}_v^k)/\tau)} \tag{2}$$
$$-\frac{1}{2} \log \frac{\exp(\text{SIM}(\bar{\boldsymbol{x}}_m^i, \bar{\boldsymbol{x}}_v^j)/\tau)}{\sum_{k=1}^M \left( \exp(\text{SIM}(\bar{\boldsymbol{x}}_m^k, \bar{\boldsymbol{x}}_v^j)/\tau) + \exp(\text{SIM}(\tilde{\boldsymbol{x}}_m^k, \bar{\boldsymbol{x}}_v^j)/\tau) + \exp(\text{SIM}(\hat{\boldsymbol{x}}_m^k, \bar{\boldsymbol{x}}_v^j)/\tau) \right)}$$

### 3.3 FLOW-MATCHING BASED MUSIC GENERATION

Flow-matching (Lipman et al., 2022) models the velocity field of transport probability path from a noise distribution $\boldsymbol{z}_0 \sim \pi_0$ to a target data distribution $\boldsymbol{z}_1 \sim \pi_1$, which is further modeled as a time-dependent changing process of probability density (a.k.a. flow), determined by the ODE $d\boldsymbol{z} = \boldsymbol{u}(\boldsymbol{z}_t|\boldsymbol{c})dt, t \in [0, 1]$, where $t$ is the time position, $\boldsymbol{z}_t$ is a point on the trajectory at time $t$, $\boldsymbol{c}$ is the condition, and $\boldsymbol{u}$ is the velocity (or the drift force). Our goal is to learn a velocity field $\boldsymbol{v}(\boldsymbol{z}_t|\boldsymbol{c};\theta)$ that approximates $\boldsymbol{u}$. Following Rectified Flow-matching (RFM) (Liu, 2022; Liu et al., 2022), we implement the target velocity field by linear interpolation between $\boldsymbol{z}_0$ and $\boldsymbol{z}_1$, leading to the RFM objective:

$$\mathcal{L}_{\text{RFM}} = \mathbb{E}_{\boldsymbol{z}_0 \sim \pi_0, \boldsymbol{z}_1 \sim \pi_1} \left[ \int_0^1 \|(\boldsymbol{z}_1 - \boldsymbol{z}_0) - \boldsymbol{v}(\boldsymbol{z}_t|\boldsymbol{c};\theta)\|^2 dt \right] \tag{3}$$

In our case, $\boldsymbol{c}$ is the visual representation, and $\boldsymbol{z}$ is the target music representation. Following the latent diffusion models (Rombach et al., 2022) and diffusion Transformers (DiT) (Peebles & Xie, 2023), we utilize a 1D variational autoencoder (VAE) (Kingma, 2013) to compress the music Mel-spectrogram into a latent representation $\boldsymbol{z} \in \mathbb{R}^{N \times C}$, reducing the computational burden of the generator, which is a DiT composed of a feed-forward transformer (FFT) and some auxiliary layers. The architecture of the DiT is shown in Figure 1.

We utilize a pre-trained unconditional DiT generator to leverage its inherent generative capabilities. The unconditional DiT shares the exact architecture, except that a learnable embedding $\varnothing \in \mathbb{R}^C$ is used to replace the visual condition $\boldsymbol{c}$. We repeat $\varnothing$ for $N$ times along the time axis to achieve this replacement. During the conditional training, the condition $\boldsymbol{c}$, the sampled point $\boldsymbol{z}_t$, and the time $t$ are transformed into the same dimension and summed element-wise. We believe that this element-wise summation (or channel-wise fusion) is crucial, as it implies point-to-point precise alignment. The fused representation is fed into the FFT blocks to estimate the velocity field. It is worth mentioning that the unconditional condition $\varnothing$ is also used to implement the generation with classifier-free guidance (CFG) (Ho & Salimans, 2022).

## 4 EXPERIMENTS

### 4.1 EXPERIMENTAL SETUP

**Data** To train the unconditional music generator, the VAE, and the vocoder, we collect a music-only dataset. We use about 50K tracks of the train split of MTG-Jamendo Dataset (Bogdanov et al., 2019), combined with 33.7K music tracks from the internet, resulting in a total of 5.3K hours of

music. We randomly segment the long tracks with a minimum and maximum duration of 4 and 30 seconds, resulting in around 1.4M music clips. We set aside about 1 hour each for test and validation. For contrastive music-visual learning and video-to-music generation, we collect 1.6K videos with semantically and rhythmically synchronized music soundtracks, resulting in around 280 hours after preprocessing. Instead of segmenting the videos, we randomly sample video segments with a minimum and maximum duration of 4 and 30 seconds within each video file during the training stage. We set aside 1 hour each for test and validation. The vocals of all the music tracks mentioned above are removed using a music source separation tool (Anjok07 & aufr33, 2020). More details are listed in Appendix D.

**Implementation** The sample rate of waveforms is 24 kHz, while the stride and number of frequency bins of the Mel-spectrograms are 300 and 160. Therefore, the VAE encoder compresses a one-second music clip into 10 frames with 8 channels. To provide a substantial amount of visual information, we sample the video at a rate of 10 FPS (which is further discussed in Appendix F). The DiT consists of 12 layers, a hidden dimension of 1024, and 16 heads. RoPE positional encoding (Su et al., 2023) and RMS normalization (Zhang & Sennrich, 2019) are also implemented. The amount of parameters of the DiT reaches 150M. We pre-train the DiT with the unconditional music dataset for 160K steps with a batch size of 1.2M frames. We perform contrastive training with the video-music dataset for 2M steps with a batch size of 160 samples, where the learnable temperature parameter is initialized as 0.07, following (Radford et al., 2021). The DiT is then trained for 60K steps with a batch size of 32. All the training is conducted using 8 NVIDIA V100 GPUs and the AdamW optimizer with a learning rate of 5e-5. More details are listed in Appendix A.

**Evaluation** For objective evaluation, we compute Frechet audio distance (FAD), Kullback–Leibler divergence (KL), inception score (IS), Frechet distance (FD), following previous music generation methods (Copet et al., 2024). To evaluate beat synchronization, we follow (Zhu et al., 2022) and compute Beats Coverage Score (BCS) and Beats Hit Score (BHS), where the former measures the ratio of overall generated beats to the total musical beats, and the latter measures the ratio of aligned beats to the total musical beats. We utilize a dynamic beat tracking algorithm (Ellis, 2007) to obtain a time-varying tempo sequence for evaluation, where the sliding window size is 8 seconds, the stride is 512 frames, and the BPM range is from 30 to 240. Also, for semantic synchronization, we use the SIM measure derived from the contrastively pre-trained encoders as a reference-free metric, where the visual encoder is selected as VideoMAE V2(base-patch16) combined with Softmax aggregation, and this selection is discussed in details in Section 4.2. The results of BCS, BHS, and SIM are measured on a percentage scale. We run inference on each test sample for 5 times and compute the average results. For subjective evaluation, we conduct crowd-sourced human evaluations with 1-5 Likert scales and report mean-opinion-scores (MOS-Q) audio quality and content alignment (MOS-A) with 95% confidence intervals. The raters are required to focus more on fine-grained aligment (such as emotion transition) in the MOS-A test. More details of the evaluation are listed in Appendix E.

**Baselines** We choose M$^2$UGen (Hussain et al., 2023) as a strong available baseline to compare. M$^2$UGen leverages an LLM (Touvron et al., 2023b) to comprehend the input information extracted from multiple multi-modal encoders (including video (Arnab et al., 2021)) and generate content-correlated music soundtracks. VidMuse (Tian et al., 2024) and V2Meow (Su et al., 2024) are also good baselines, but they have not open-sourced their training or inference code. Other works, such as CMT (Di et al., 2021) and D2M-GAN (Zhu et al., 2022) are not considered for comparison because their scope of application differs from ours (symbolic music generation or dance-to-music), which could result in an unfair comparison. To provide a more comprehensive perspective, we construct an additional weak baseline for comparison: MuVi(beta), where we use CLIP-ViT(base-patch16) and the attention pooling adaptor as the visual encoder, and the whole model (adaptor and generator) is trained from scratch (without any pre-training) with the video dataset. For the proposed model, we use VideoMAE V2(base-patch16) and Softmax aggregation as the visual encoder for most of the comparison (this selection is discussed in Section 4.2).

### 4.2 RESULTS OF VIDEO-TO-MUSIC GENERATION

**Analysis of Visual Representation Modeling Strategies** We compare different visual encoders combined with the corresponding visual adaptor strategies. The results are listed in Table 1. To investigate the impact of temporal resolution, we extract the CAVP features from videos that are downsampled into two different frame rates, 4 FPS and 10 FPS, where Luo et al. (2024) originally

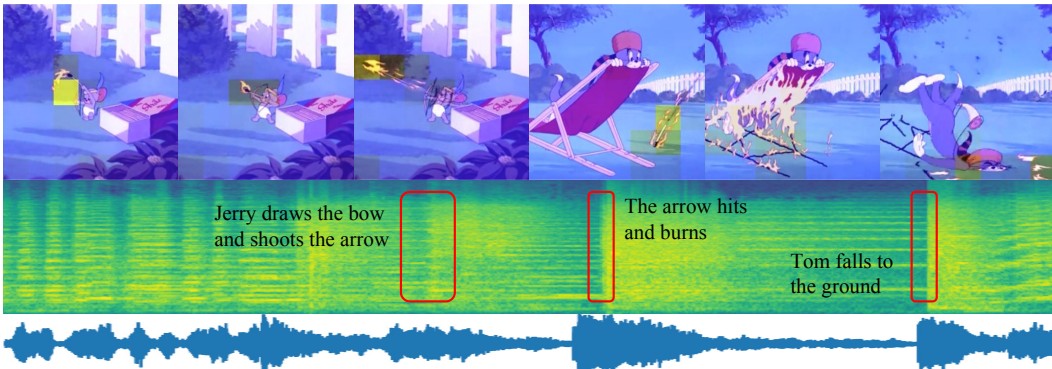

Figure 2: Visualization of the attention distribution of Softmax aggregation. The yellower the patch, the more it is related to the generated music. We mask the video frames with the averaged attention scores. We transform the patches corresponding to the weights after applying Softmax into masks, and then adjust the colors of the masks accordingly. When the weights are smaller (close to 0.0), the mask appears bluer; conversely (close to 1.0), it appears yellower. This reflects the attention distribution of the adaptor.

Table 1: Results of different visual encoders and adaptors. The **bold numbers** represent the best result of that column, and the underlined numbers represent the second best. "Softmax" and "Sigmoid" represent the Softmax and Sigmoid aggregation strategies, "Attention" means the attention pooling strategy, and "Average" and "CLS" indicate average pooling and pooling with the CLS token.

| Visual Encoder | Adaptor | FAD↓ | KL↓ | IS↑ | FD↓ | BCS↑ | BHS↑ | SIM↑ | MOS-Q↑ | MOS-A↑ |
|---|---|---|---|---|---|---|---|---|---|---|
| CAVP (4 FPS) | - | 5.45 | 4.13 | 1.45 | 38.95 | 87.50 | 41.90 | 4.05 | 3.59±0.06 | 3.41±0.10 |
| CAVP (10 FPS) | - | 5.31 | 4.05 | 1.39 | 37.04 | 88.93 | 45.18 | 4.08 | 3.67±0.07 | 3.45±0.12 |
| | Softmax | 5.12 | 3.53 | 1.62 | 31.28 | **106.33** | **50.05** | 15.38 | 3.71±0.06 | 4.14±0.10 |
| | Sigmoid | 6.01 | 3.85 | 1.66 | 28.94 | 102.19 | 48.95 | 14.62 | 3.63±0.05 | 4.03±0.06 |
| CLIP | Attention | 4.36 | **3.46** | 1.51 | 28.55 | 105.23 | 49.81 | 16.35 | 3.77±0.03 | 4.13±0.05 |
| | Average | 6.40 | 3.76 | 1.61 | 32.79 | 105.45 | 49.52 | 14.82 | 3.54±0.07 | 3.94±0.09 |
| | CLS | 7.40 | 4.04 | **1.71** | 36.32 | 103.06 | 49.50 | 13.47 | 3.56±0.08 | 3.89±0.06 |
| | Softmax | 4.93 | 3.80 | 1.43 | 30.71 | 101.15 | 48.87 | 18.47 | 3.73±0.06 | 4.12±0.08 |
| VideoMAE | Sigmoid | 4.36 | 3.56 | 1.44 | 31.06 | 99.16 | 47.94 | 16.53 | 3.74±0.07 | 4.06±0.04 |
| | Average | 5.13 | 4.02 | 1.37 | 33.78 | 97.82 | 47.01 | 15.59 | 3.63±0.06 | 4.02±0.09 |
| | Softmax | **4.28** | 3.52 | 1.63 | **28.15** | 104.17 | 49.23 | **19.18** | **3.81±0.05** | **4.15±0.08** |
| VideoMAE V2 | Sigmoid | 4.89 | 3.72 | 1.41 | 33.21 | 101.35 | 48.88 | 18.82 | 3.75±0.09 | 4.12±0.07 |
| | Average | 4.75 | 4.04 | 1.55 | 31.71 | 99.85 | 48.28 | 15.67 | 3.52±0.12 | 3.96±0.09 |

uses the former. Clearly, increasing the frame rate helps improve synchronization, but the benefit is limited. This may be because CAVP features are primarily trained with video and monotonous sound pairs, lacking information about music. All other methods are trained using the two-stage training process. All the pre-trained visual encoders are base and patch-16 version. From the results, we found that both CLIP+Attention and VideoMAE V2+Softmax methods are very competitive, but we ultimately choose the latter for further experiments due to its slight overall advantage. Moreover, it utilizes unsupervised training with video data, which may have potential knowledge about the temporal dimension of videos. In addition, we explore whether the Softmax aggregation functions effectively, and we visualize the global Softmax weights in Figure 2, where we can observe that the adaptor indeed focuses on more critical and rapidly changing local features. However, considering this is the Softmax aggregation strategy, not the attention pooling with a CLS token as the query, we can also view the gating function $g(\cdot)$ as a learnable query to form global attention.

**Comparison with the Baselines**   The results of comparison of different systems are illustrated in Table 2. We directly use the model checkpoints of M²UGen provided by Hussain et al. (2023) to generate music soundtracks. As M²UGen does not have any specific designs or measures for synchronization, its performances in terms of BCS, BHS, and SIM are relatively poor. However, M²UGen has a strong music generator, MusicGen (Copet et al., 2024), combined with a powerful

Table 2: Results of several V2M systems.

| Method | FAD↓ | KL↓ | IS↑ | FD↓ | BCS↑ | BHS↑ | SIM↑ | MOS-Q↑ | MOS-A↑ |
|---|---|---|---|---|---|---|---|---|---|
| MuVi(beta) | 4.56 | 4.25 | 1.54 | 35.19 | 95.21 | 45.19 | 10.71 | 3.55±0.08 | 3.89±0.05 |
| M²UGen | 5.12 | **3.83** | **1.65** | 32.14 | 75.21 | 25.14 | 1.41 | 3.79±0.09 | 3.19±0.14 |
| MuVi | **4.28** | **3.52** | 1.63 | **28.15** | **104.17** | **49.23** | **19.18** | **3.81±0.05** | **4.15±0.08** |

Table 3: Results of in-context learning (ICL).

| Method | CLAP | FAD↓ | KL↓ | IS↑ | FD↓ | BCS↑ | BHS↑ | SIM↑ | MOS-Q↑ | MOS-A↑ |
|---|---|---|---|---|---|---|---|---|---|---|
| MuVi | - | 4.28 | 3.52 | 1.63 | 28.15 | 104.17 | 49.23 | 19.18 | 3.81±0.05 | 4.15±0.08 |
| MuVi (ICL) | 0.35 | 6.13 | 3.71 | 1.48 | 34.16 | 93.30 | 44.06 | 13.55 | 3.84±0.07 | 3.95±0.09 |

prompt generation LLM, ensuring overall music quality. MuVi(beta) is trained without any pre-trained models, namely, the adaptor and the DiT are trained from random initialization with only the video data, resulting in lower sound quality, diversity, and poorer synchronization. However, the channel-wise fusion of the visual conditioning still aids in synchronization, ensuring that the alignment does not fall below an acceptable level.

## 4.3 EXPLORING MUSIC GENERATION WITH IN-CONTEXT LEARNING

We further investigate the in-context learning capability of MuVi, which enables us to control the style of generated music by a given prompt. Specifically, when training the unconditional DiT, we apply the partial denoising technique (Gong et al., 2022) – that is, only part of the latent sequence is used to compute the trajectory interpolation $z_t$, while the other part (i.e., the context) remains clean. The loss is not applied to the context area. We illustrate the mechanism in Figure 3. Therefore, the previously unconditional DiT is now conditional, dependent on the given music prompt. We also randomly apply this partial denoising mechanism during training with a probability of 0.8 to enable CFG. We randomly crop 1∼4 seconds or 33% of the original length of the waveform at the beginning to form the context, depending on which one is smaller. During the V2M training stage, the visual condition $c$ is also segmented accordingly because the context needs no condition. To evaluate the performance

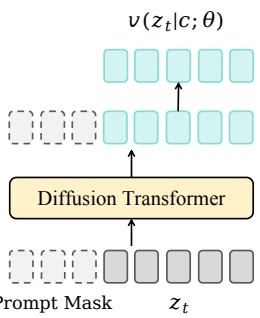

Figure 3: Illustration of In-context Learning.

of in-context learning, we use CLAP (Elizalde et al., 2023) to compute the overall cosine similarity between the music prompt and the generated soundtrack. The prompt soundtracks are sampled from the test split of the music-only dataset. The results are listed in Table 3, where we can see that the involvement of context degrades the performance, in terms of both audio quality and synchronization. However, the subjective evaluation results reveal that the overall performance is still satisfactory, demonstrating the potential style-controlling capability of MuVi.

## 4.4 ABLATION STUDY

**Analysis of Contrastive Pre-training** We analyze the effectiveness of the proposed contrastive music-visual pre-training strategy by comparing different pre-training settings. We evaluate the performances when dropping the contrastive pre-training phase, conducting only the basic contrastive learning, and involving two additional negative sampling strategies. The results are listed in Table 4, from which we can observe the performance decline when removing contrastive pre-training procedures. Interestingly, introducing contrastive pre-training degrades the performance in terms of FAD. However, there may be a reasonable explanation for this when viewed globally: the KL divergence is relatively high when dropping the contrastive learning (or applying the basic one), indicating a lower diversity in generation. Lower diversity does not necessarily mean a higher FAD; sometimes, it may also cause the FAD to decrease. We believe that the absence of contrastive learning, or merely a basic version, might lead to a certain degree of overfitting, thereby reducing the diversity of the generated music. Introducing more types of negative samples implicitly creates an information bottleneck, causing the model to focus on more general features related to alignment and synchronization.

Table 4: Results of different settings of pre-training. PT(DiT) indicates whether the DiT is pre-trained unconditionally with music; CL stands for basic contrastive learning; TS and RR stand for the involvement of negative samples constructed from temporal shift and random replacement, respectively.

| PT(DiT) | CL | TS | RR | FAD↓ | KL↓ | IS↑ | FD↓ | BCS↑ | BHS↑ | SIM↑ | MOS-Q↑ | MOS-A↑ |
|---|---|---|---|---|---|---|---|---|---|---|---|---|
| × | ✓ | ✓ | ✓ | 6.82 | 4.28 | 1.33 | 32.75 | 101.83 | 49.57 | 17.15 | 3.45±0.09 | 3.93±0.06 |
| ✓ | × | × | × | **4.21** | 3.69 | 1.68 | 28.49 | 97.35 | 47.01 | 8.42 | 3.75±0.05 | 4.01±0.08 |
| ✓ | ✓ | × | × | 4.25 | 3.73 | 1.59 | 28.19 | 99.43 | 48.53 | 17.73 | 3.77±0.06 | 4.05±0.06 |
| ✓ | ✓ | ✓ | × | 4.31 | 3.65 | 1.57 | 28.36 | 103.78 | **49.45** | 18.80 | 3.80±0.09 | **4.15±0.12** |
| ✓ | ✓ | × | ✓ | 4.24 | 3.55 | 1.53 | 28.21 | 102.91 | 49.17 | 18.69 | 3.79±0.08 | 4.13±0.07 |
| ✓ | ✓ | ✓ | ✓ | 4.28 | **3.53** | **1.53** | **28.15** | **104.17** | 49.23 | **19.18** | **3.81±0.05** | **4.15±0.08** |

Table 5: Results of different model sizes.

| Model Size | FAD↓ | KL↓ | IS↑ | FD↓ | BCS↑ | BHS↑ | SIM↑ | MOS-Q↑ | MOS-A↑ |
|---|---|---|---|---|---|---|---|---|---|
| Small (85M) | 5.21 | 3.87 | 1.41 | 32.47 | 98.14 | 47.75 | 16.33 | 3.66±0.07 | 3.99±0.09 |
| Base (150M) | 4.28 | 3.52 | 1.63 | 28.15 | 104.17 | 49.23 | 19.18 | 3.81±0.05 | 4.15±0.08 |
| Large (330M) | 4.25 | 3.49 | 1.65 | 28.11 | 104.23 | 49.56 | 19.24 | 3.82±0.06 | 4.17±0.09 |

**Analysis of Model Size and Parameter Initialization** We compare different sizes of model parameters to investigate the scalability of MuVi. We label the major model as "base", and construct "small" and "large" versions by adjusting the parameter dimensions and model layers. The details of model architectures of different sizes are listed in Appendix A. The model sizes and the corresponding results are listed in Table 5. The results align with the common sense that a larger model leads to better performance. Although the performance improvement is limited after the model size increases (possibly due to the data volume not increasing correspondingly), this suggests the potential scalability to larger model size and amount of data. In addition, we compare the model performance when the music generator is not unconditionally pre-trained in advance, and we report the results in the first row of Table 4. From the results, we can observe that the audio diversity (in terms of KL) decreases, and so do the audio quality and synchronization.

**Analysis of CFG Scale** We investigate the impact of various CFG scales on the performance of MuVi, and the results are illustrated in Figure 4. Initially, the plausibility and diversity increase with the CFG scale, in terms of FAD, KL, IS, and FD, reaching an optimal value at around 3, 4, and 5. After that, the results start to drop. Regarding synchronization (BHS and SIM), the measures reach a peak and then tend to stabilize, without further obvious increases. Taking various factors into account, and primarily considering the audio quality and synchronization, we choose a CFG scale of 4.

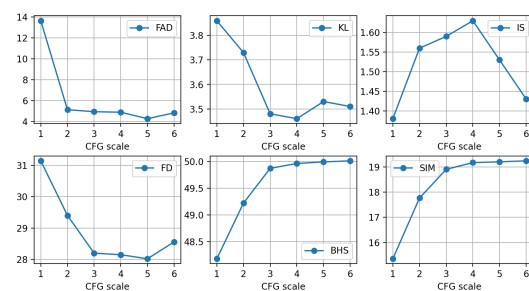

Figure 4: Results of different CFG scales.

## 5 CONCLUSION

In this paper, we introduced MuVi, a novel V2M method that generates music soundtracks with both semantic alignment and rhythmic synchronization. We leveraged a simple non-autoregressive ODE-based music generator, combined with an efficient visual adaptor that compressed visual information and ensured length alignment. An innovative contrastive music-visual pre-training scheme was constructed to emphasize temporal synchronization by addressing the periodic nature of beats. Experimental results revealed that the proposed method achieved satisfactory results in the V2M task, and we have investigated the effectiveness of different designs. For future work, we will explore controllable V2M methods with textual prompts, which generate music with styles or emotions aligned with the textual descriptions. Further discussions are provided in the appendix.

## 6 ETHICS STATEMENT

The proposed method, MuVi, is designed to advance V2M technologies. If used legitimately, this technology can benefit many applications, such as multimedia social platforms, advertising, gaming, movies, and more. However, we also acknowledge the potential risks of misuse, such as the production of pirated audio and video content. We plan to impose certain restrictions on this technology, such as regulating its use through licensing. We emphasize these ethical concerns to ensure the healthy and positive development of AI technology.

## 7 REPRODUCIBILITY STATEMENT

We take several steps to ensure the reproducibility of the experiments presented in this paper: 1) The algorithm and configuration of the contrastive music-visual pre-training are described in Section 3.2 and Section 4.1; 2) The architectures and hyperparameters of the visual adaptor, the DiT, the VAE, and the vocoder are elaborated in Section 3, Section 4.1, Appendix A, and Appendix C; 3) The evaluation metrics, including FAD, KL, IS, FD, BCS, BHS, SIM, MOS-Q, and MOS-A are described in detail in Section 4.1 and Appendix E; 4) For the unconditional music pre-training, we utilize a combination of a publicly available dataset and a web-crawled dataset, while we utilize a fully web-crawled dataset to train the V2M model, because the publicly available datasets are insufficient for our task. We describe the datasets in Appendix D. Objective results are reported based on the average performance of multiple inferences.

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

## A  ARCHITECTURE AND IMPLEMENTATION DETAILS

The main model, i.e., the velocity field estimator, is implemented as a multi-layer feed-forward Transformer. The detailed sizes and the parameter count are listed in Table 6. Following LLaMA (Touvron et al., 2023a), we adopt RoPE positional encoding (Su et al., 2023) and RMS Normalization layers (Zhang & Sennrich, 2019) in the DiT. We utilize the Flash attention technique (Dao et al., 2022) to save memory and accelerate computation. During the training stage, we apply the mixed precision training strategy. Specifically, in all places with residual connections and normalization calculations, we convert the data to `fp32`, while in other cases, we convert it to `bf16`.

As for the 1D VAE, we adapt the VAE configuration from Rombach et al. (2022) and build a 1D version. Specifically, we use 1D convolution layers instead of 2D and drop the attention layers to support arbitrary-length input. The channel dimension is 256 with channel multipliers [1, 2, 4, 8], while the latent dimension is set to 8. Therefore, the VAE model compresses 1-second audio into 10 latent frames. We train the VAE model with a batch size of 640K steps for 350K steps.

As for the vocoder, we utilize a HiFi-GAN (Kong et al., 2020a) generator with 4 upsample layers, where the upsample rates are [5, 5, 4, 3], resulting in the stride of 300. The upsample kernel sizes are [16, 8, 8, 4], and the initial channel dimension for upsampling is 1200. We train the vocoder with a batch size of 102.4K frames for 520K steps. The multi-period discriminator (MPD) and the multi-scale discriminator (MSD) are adopted to improve generation quality.

As for the visual adaptor, we use a single 2D convolution layer with a stride of $2 \times 2$ to downsample the output features from visual encoders. Specifically, if the encoder output is $z \in \mathbb{R}^{N \times (H' \times W') \times C}$ (will be explained in Appendix C), the numbers of patches along the two directions, $H'$ and $W'$, should be even numbered. For example, if we use VideoMAE as the visual encoder, then $H' = 224/16 = 14$. Therefore, the 2D convolution layer will further downsample the feature to $z \in \mathbb{R}^{N \times (\frac{H'}{2} \times \frac{W'}{2}) \times C}$. After this downsampling, we then apply various compression strategies, followed by a linear layer for output:

- **Gated Aggregation.** We utilize a single-layer convolution layer and a nonlinear function to obtain the dynamic weights. The convolution layer has a kernel size of 3 and 4 filters, so the output dimension is 4, and 4 groups of weights are generated. We then compute 4 weighted averages on the downsampled features with the 4 groups of weights before computing the average of groups. If the nonlinear function is Sigmoid, the average of groups will be further divided by the average of weights to ensure that the sum of all weights within one group is 1.

- **Attention Pooling.** We utilize three linear layers to transform the downsampled features to query, key, and value. When computing the attention weights, we apply multi-head attention (Vaswani, 2017) by splitting the query and key vectors into 4 segments, and compute the attention scores within each segmented group. The values are also segmented into heads, which are eventually concatenated along the channel dimension before the output layer.

- **Vanilla Pooling.** For average pooling, we simply compute the average of the downsampled visual features before the output layer. For CLS pooling, we directly feed the CLS embeddings into the output layer.

Table 6: Model configurations of the DiT with different sizes.

| Hyperparameter | Small | Base | Large |
|---|---|---|---|
| Hidden dimension | 768 | 1024 | 1280 |
| #Layers | 12 | 12 | 16 |
| #Attention heads | 16 | 16 | 16 |
| #Parameters | 85M | 150M | 330M |

Note that all the hidden dimensions mentioned in the visual adaptor are equal to that of the DiT.

## B   DETAILS OF TRAINING AND INFERENCE

**Music Representation**   Given the Mel-spectrogram of a music clip, we utilize an 1-D VAE encoder to compress it into a latent representation $z \in \mathbb{R}^{N \times C}$, with a temporal compression ratio of 8 and a channel compression ratio of 20. A KL penalty $0.01$ is implemented (Rombach et al., 2022). A HiFi-GAN vocoder (Kong et al., 2020a) is utilized to recover the decoded spectrograms to waveforms.

**Video Representation**   We follow the corresponding pre-trained visual encoder to preprocess the videos. We preserve the temporal dimension of the hidden states of encoder outputs to support fine-grained alignment. More details of the inference procedure of the pre-trained encoders can be found in Appendix C.

**Training Procedure**   The training procedure of the main model is divided into two stages: 1) Given the pre-trained visual encoder, we pre-train the visual adaptor with the auxiliary audio encoder with the contrastive objective $\mathcal{L}_T$, where only the visual adaptor and the head of the audio encoder are learnable; 2) Given the unconditionally pre-trained DiT with the flow-matching objective $\mathcal{L}_{\text{RFM}}$, we further train it with the V2M task where the visual adaptor and the DiT are learnable. During training, we randomly replace the visual condition $c$ with the unconditional condition $\varnothing$ with a probability of 0.2 to enable CFG in inference.

**Inference Procedure**   The input video is preprocessed according to the selected visual encoder, and then compressed into the visual condition with the cascaded visual encoder and adaptor. We adopt the Euler sampler with a fixed step size to solve the ODE trajectory. If not otherwise stated, we use 25 sampling steps for generation. To enable CFG, a modified velocity field estimate is implemented: $v_{\text{CFG}}(z_t|c;\theta) = \gamma v(z_t|c;\theta) + (1-\gamma)v(z_t|\varnothing;\theta)$, where $\gamma$ is the guidance scale trading off the sample diversity and generation quality. If not otherwise stated, $\gamma$ is set to 4, which is further discussed in Section 4.4.

## C   DETAILS OF PRE-TRAINED ENCODERS

Most video encoders (VideoMAE, VideoMAE V2, etc.) and audio encoders (AudioMAE, PANNs (Kong et al., 2020b), etc.) only support fixed-length signal input, as they are widely used in understanding or classification tasks. However, since our method focuses on temporal alignment and synchronization, and the input is of variable length, we need to modify the inference procedure of these models to recover the temporal dimension of the features.

**VideoMAE and VideoMAE V2**   Given a batch of preprocessed videos $x \in \mathbb{R}^{B \times N \times 3 \times H \times W}$ during training, where $B$ is the batch size, $N$ is the varying temporal length (number of frames), 3 indicates three color spaces, $H$ and $W$ are the size of the image, we segment the frames along the temporal dimensions with a segment length of $L$, where $L$ is the only supported frame length of the visual encoder. In this case, $L = 16$ for both VideoMAE and VideoMAE V2. It is quite likely that $L$ cannot evenly divide $N$, that is, $\lfloor N/L \rfloor \times L \neq N$. If so, we stack the first $M$ segments together on the batch size dimension to obtain $\dot{x} \in \mathbb{R}^{(B \times M) \times L \times 3 \times H \times W}$, where $M = \lfloor N/L \rfloor$, leaving out $N - M \times L$ frames. Now, instead of padding the remaining frames with zeros to $L$ frames, we fetch the last contiguous $L$ frames of the video clip and append it to $\dot{x}$ to obtain $\ddot{x} \in \mathbb{R}^{(B \times M+1) \times L \times 3 \times H \times W}$. Then

we run the visual encoders to have the last hidden states $\hat{z} \in \mathbb{R}^{(B \times M+1) \times (L' \times H' \times W') \times C}$ as outputs, where $L'$ is the compressed temporal dimension, $H'$ and $W'$ are the number of patches along two directions, and $C$ is the channel dimension. In this case, $L' = L/2$, $H' = H'/16$, and $W' = W'/16$, since the compression ratio of the visual encoders is $2 \times 16 \times 16$ (which also requires that $N$ should be an even number, leading to a compression of $N' = N/2$). Then we unpack $B \times M + 1$ segments and connect each other end-to-end to recover $B$ batches of $N$ frames. Note that we only recover the last $N' - M \times L'$ frames of the last segment. The resulting representation $z \in \mathbb{R}^{B \times N' \times (H' \times W') \times C}$ is eventually fed to the subsequent modules.

**AudioMAE**   We solve the varying-length problem of the audio encoder with the same strategy as visual encoders. We use a fixed window size of 10 seconds (supported by AudioMAE) to segment the input waveform, filling continuous values ahead for the last incomplete segment. We then recover the feature with varying temporal lengths and crop the last potential incomplete segment.

# D   DETAILS OF DATA

We crawl 1.6K videos with semantically and rhythmically synchronized music soundtracks from the internet. These videos are mostly artistic creations like movies and TV series, which differ from other music videos in that their music is also part of the creation. In other words, while other videos might be created first and then set to music, the creative process for these videos often involves simultaneous creation of both music and video, or even music being composed first followed by video production based on the rhythm and melody. Therefore, these videos generally surpass ordinary music videos in terms of audio-visual synchronization, music quality, and video quality. We referenced a variety of data sources, such as Walt Disney (Disney, 1940), Tom and Jerry (Hanna & Barbera, 1940), Charlie Chaplin comedies, and more. These masterpieces provided a high-quality data foundation for our work.

Specifically, we collect 1.6K videos, ranging in duration from less than 1 minute to hours. Instead of segmenting the videos beforehand, we adopt a continuous random sampling strategy during training. That is, for a batch of videos during training, we sample a random segment of each video on the fly. This strategy, compared to segmentation beforehand, allows more diversity in data distribution and forms a data enhancement to some extent. However, because the duration of each video largely differs, we apply a data sampler based on the video duration to balance the sampling frequency. It is also worth mentioning that this strategy ensures that all the video clips within one batch come from different videos, which further ensures the effectiveness of the negative samples in the contrastive learning.

# E   DETAILS OF EVALUATION

## E.1   OBJECTIVE EVALUATION

We utilize the audio evaluation toolkit provided by Liu et al. (2023). The FAD scores are computed from the embedding statistics of a VGGish classifier (Hershey et al., 2017), while the FD scores are computed using PANNs (Kong et al., 2020b). For BHS, we compute the alignment between the ground-truth beat sequences and the generated based on a 100ms tolerance. This means that if the time difference between a predicted beat and a real beat is less than 100ms, then the predicted beat is considered correct, and they will be paired (or aligned). These pairings are exclusive. We obtain the alignment between two sequences by constructing a prediction-reference pairing graph and solving the bipartite matching problem. We utilize a dynamic programming-based beat tracking algorithm (Ellis, 2007) to compute time-varying tempo sequences from both ground truth and generation. The highest accuracy reported for this method is 58.8% in their initial paper. We choose this method due to its simplicity and efficiency, and the metrics obtained could be considered as a reference.

## E.2   SUBJECTIVE EVALUATION

For each task, 20 samples are randomly selected from our test set for subjective evaluation. Professional listeners, totaling 20 individuals, are engaged to assess the performance. In MOS-Q evaluations, the focus is on overall generation quality, encompassing the melodic nature and sound quality. For

Table 7: Results of different video frame rates.

| FPS | FAD↓ | KL↓ | IS↑ | FD↓ | BCS↑ | BHS↑ | SIM↑ | MOS-Q↑ | MOS-A↑ |
|---|---|---|---|---|---|---|---|---|---|
| 2 | 7.86 | 4.24 | 1.28 | 39.18 | 85.33 | 40.12 | 11.12 | 3.71±0.03 | 3.76±0.07 |
| 4 | 5.38 | 4.14 | 1.39 | 34.24 | 96.46 | 47.13 | 15.26 | 3.75±0.07 | 3.93±0.05 |
| 10 | 4.28 | 3.52 | 1.63 | 28.15 | 104.17 | 49.23 | 19.18 | 3.81±0.05 | 4.15±0.08 |
| 20 | 4.25 | 3.53 | 1.65 | 28.11 | 104.22 | 50.08 | 20.11 | 3.83±0.08 | 4.16±0.05 |

MOS-A, listeners are required to focus on semantic alignment and rhythmic synchronization, disregarding audio quality. In both MOS-Q and MOS-A evaluations, participants rate various music samples on a Likert scale from 1 to 5. Participants were duly informed that the data were for scientific research use.

# F    EXTENSIONAL EXPERIMENTS

## F.1    ANALYSIS OF VIDEO FRAME RATE

We compare the impact of different video frame rates. Specifically, we select a different frame rate to sample the frame sequence, which is then processed by the VideoMAE V2 encoder as mentioned in Appendix C. To align with the music representation, we resample the visual features to meet the target length with linear interpolation. The results are listed in Table 7. The improvement in frame rate clearly enhances the performance when the rate is lower. However, as the frame rate reaches a certain level, the improvement in performance gradually becomes limited. To balance effectiveness and computational load, we chose a frame rate of 10 FPS.

