# OpenReview forum: "MuVi: Video-to-Music Generation with Semantic Alignment and Rhythmic Synchronization"
_ICLR.cc/2025/Conference — Submitted to ICLR 2025_

### Official Review · Reviewer_yVkt · 2024-11-03

**Soundness:** 2
**Presentation:** 2
**Contribution:** 2
**Rating:** 3
**Confidence:** 5

**Summary:**

This paper proposes MuVi, a new method for generating music that aligns with video content, focusing on both semantic alignment and rhythmic synchronization. MuVi's design includes a "visual adaptor" that extracts relevant visual features from videos, which helps guide music generation to match the mood and rhythm of the video. To improve synchronization between visual events and musical beats, the authors use a pre-training technique that contrasts synchronized and unsynchronized video-music pairs, helping the model learn rhythmic alignment.

**Strengths:**

1. This paper introduces a new method for generating music that aligns with video content, focusing on both semantic alignment and rhythmic synchronization within a generative video-music Diffusion Transformer framework.
2. The model employs a joint encoder-decoder architecture that integrates a contrastive pre-training scheme for improved synchronization. The inclusion of a "visual adaptor" enhances the model’s ability to compress and process high-frame-rate visual inputs, capturing video cues for music generation.
3. The paper is well-organized, presenting MUVI's methodology alongside a series of experiments. The framework demonstrates superior performance over the baseline on the test dataset across various evaluation metrics, showcasing its effectiveness in video-to-music generation.

**Weaknesses:**

1.Novelty and Contribution: The paper presents its main contributions as a visual adaptor and a contrastive training scheme, but visual adaptor techniques and contrastive learning have already been used in video-to-music generation tasks [1, 2] and are commonly employed in multi-modal learning [3, 4]. The design of the visual adaptor lacks unique innovation, primarily involving a selection of common aggregation and pooling methods, which appears more as an ablation study to find the best setting. Overall, the proposed method lacks novelty, and the results in Table 2 indicate that the proposed method does not outperform the baseline across all metrics.

2.Lack of Justification and Explanation: Another weakness is the lack of clear justification and explanation across different sections, from design choices to metric selection.

2.1 The adaptor design section lacks a clear justification. For instance, why were these three adaptor methods chosen, instead of exploring alternative multi-modal adaptors [1, 3]? Why is CLS set as the query instead of the key-value pair?

2.2 The paper introduces various metrics for evaluating model performance, but lacks explanations for each metric. For instance, in lines 446-449: “resulting in lower sound quality, diversity, and poorer synchronization,” it is unclear which metrics specifically measure sound quality, diversity, or synchronization. Additionally, the statement “the channel-wise fusion of the visual conditioning still aids in synchronization” lacks experimental evidence to substantiate this claim.

2.3 Ambiguous phrases like “we believe” (lines 222, 483) and “might lead to” (lines 77, 483) appear multiple times in the paper. Clear support or reasoning should be provided for these assertions.

3. Presentation and Writing: There are some presentation and writing issues within the paper.

3.1 The introduction (line 63) highlights tackling “Integration of foley and sound 	effects,” yet no further details or experiments addressing this topic are provided in the rest of the paper.

3.2 The temporal shift method introduced in Section 3.2 is motivated as a significant contribution, but it lacks a clear explanation. Additionally, some symbols, like "m," are redefined multiple times, "C'" is not defined, which may cause confusion for readers.


4. Experimental Comparison: A main weakness of the paper is lack of the experimental comparisons, which include only one baseline method.

4.1 A simple baseline could have been constructed by combining an existing video understanding model with a music generation model, similar to the approach in [2, 6].

4.2 The experiments omit comparisons with several relevant state-of-the-art methods, such as Diff-BGM [5], VidMuse [6], and Dance2Music-Diffusion [7].

4.3 The M^2Ugen method shows comparable or superior results in terms of audio quality (Table 2). Fine-tuning this method on the dataset used in this paper could provide additional insight into its performance.

References:
[1] Liu S, Hussain A S, Sun C, et al. M$^{2}$UGen: Multi-modal Music Understanding and Generation with the Power of Large Language Models[J]. arXiv preprint arXiv:2311.11255, 2023.

[2] Lin Y B, Tian Y, Yang L, et al. VMAS: Video-to-Music Generation via Semantic Alignment in Web Music Videos[J]. arXiv preprint arXiv:2409.07450, 2024.

[3] Zhang R, Han J, Liu C, et al. Llama-adapter: Efficient fine-tuning of language models with zero-init attention[J]. arXiv preprint arXiv:2303.16199, 2023.

[4] Radford A, Kim J W, Hallacy C, et al. Learning transferable visual models from natural language supervision[C]//International conference on machine learning. PMLR, 2021: 8748-8763.

[5] Li S, Qin Y, Zheng M, et al. Diff-BGM: A Diffusion Model for Video Background Music Generation[C]//Proceedings of the IEEE/CVF Conference on Computer Vision and Pattern Recognition. 2024: 27348-27357.

[6] Tian Z, Liu Z, Yuan R, et al. VidMuse: A Simple Video-to-Music Generation Framework with Long-Short-Term Modeling[J]. arXiv preprint arXiv:2406.04321, 2024.

[7] Zhang C, Hua Y. Dance2Music-Diffusion: leveraging latent diffusion models for music generation from dance videos[J]. EURASIP Journal on Audio, Speech, and Music Processing, 2024, 2024(1): 48.

**Questions:**

In addition to the weaknesses, here are some points that raise further confusion or seem inconsistent in the paper:

1.Claim on Previous V2M Methods (lines 39-41): The authors claim that "Previous V2M methods focus on global features," presenting this as a limitation of past approaches. However, this appears inconsistent with prior work, as several existing methods focus on local clip features for training. For instance, V2Meow [1] and VMAS [2] emphasize local clip-based features, while VidMuse [3] captures both local and global features through long-short-term modeling. The authors should clarify and provide evidence to support their assertion about the emphasis on global features in previous V2M approaches.

2.Choice of Beat Synchronization Metrics and Exclusion of Dance Video Music Generation for Comparison: The authors select Beats Coverage Score (BCS) and Beats Hit Score (BHS) as metrics to evaluate beat synchronization, following the approach in [4] (line 346), which specifically targets music generation for dance videos. However, the authors then claim in line 364 that "D2M-GAN are not considered for comparison because their scope of application differs from ours." If dance-related videos are outside MuVi’s intended scope, it is unclear why dance-specific metrics are being applied for evaluation. This raises a need for clarification.

3.Choice of MuVi(beta) Setting for Comparison: The paper claims "use CLIP-ViT(base-patch16) and the attention pooling adaptor as the visual encoder" for MuVi(beta) (lines 366-367). However, Table 1 shows that the VideoMAE V2 with a Softmax adaptor yields better results for this setting. It is unclear why a suboptimal setting was selected for MuVi(beta), as this choice could impact the fairness and interpretability of the comparisons. An explanation from the authors on the rationale for this choice would provide more clarity.

[1] Su K, Li J Y, Huang Q, et al. V2Meow: Meowing to the Visual Beat via Video-to-Music Generation[C]//Proceedings of the AAAI Conference on Artificial Intelligence. 2024, 38(5): 4952-4960.

[2] Lin Y B, Tian Y, Yang L, et al. VMAS: Video-to-Music Generation via Semantic Alignment in Web Music Videos[J]. arXiv preprint arXiv:2409.07450, 2024.

[3] Tian Z, Liu Z, Yuan R, et al. VidMuse: A Simple Video-to-Music Generation Framework with Long-Short-Term Modeling[J]. arXiv preprint arXiv:2406.04321, 2024.

[4] Zhu Y, Olszewski K, Wu Y, et al. Quantized gan for complex music generation from dance videos[C]//European Conference on Computer Vision. Cham: Springer Nature Switzerland, 2022: 182-199.

---

> ### Author Response · Authors · 2024-11-21
> **Response to Reviewer yVkt (Part 1/N)**
>
> Thank you for your valuable comments, and we would like to make some clarifications, which we hope will address your concerns.
>
> ### **Novelty and Contribution**
>
> > 1.Novelty and Contribution: The paper presents its main contributions as a visual adaptor and a contrastive training scheme, but visual adaptor techniques and contrastive learning have already been used in video-to-music generation tasks [1, 2] and are commonly employed in multi-modal learning [3, 4]. The design of the visual adaptor lacks unique innovation, primarily involving a selection of common aggregation and pooling methods, which appears more as an ablation study to find the best setting. Overall, the proposed method lacks novelty, and the results in Table 2 indicate that the proposed method does not outperform the baseline across all metrics.
>
> We disagree with reviewer yVkt's opinion that our method lacks innovation.
>
> 1. Foremost, prior to our work, there were no studies that simultaneously focused on general video-to-music generation with semantic alignment and rhythmic synchronization. We identified and addressed this issue, achieving significantly superior results. Therefore, we are tackling a new task.
> 2. Indeed, we adopted two kinds of widely used methods: visual adaptors and contrastive learning. However, our visual adaptor is designed to compress high-frame-rate video features and provide a non-autoregressive generator with features about video semantics and synchronization. For contrastive learning, we designed two novel types of negative samples for the frame-level contrastive learning scheme, and as far as we know, none of the methods mentioned by the reviewer yVkt [1, 2, 3] uses frame-level contrastive learning scheme nor introduces new negative samples.
> 3. The reviewer yVkt claims that our method's failure to outperform M$^2$UGen in terms of IS detracts from its novelty, according to Table 2. However, M$^2$UGen, from the perspective of music generation, is a strong baseline, because it incorporates a strong music generator, MusicGen. It is challenging for our method to completely surpass a model that has ten times our parameter count and four times our training data volume, especially considering that our focus is not solely on music generation itself. The ultimate quality of music generation is not the focus of this paper; it only needs to be satisfactory. The emphasis of this paper is on the semantic alignment and rhythmic synchronization.

---

> > ### Comment · Reviewer_yVkt · 2024-11-25
> >
> > (1) The authors state, "Foremost, prior to our work, there were no studies that simultaneously focused on general video-to-music generation with semantic alignment and rhythmic synchronization."
> > This is not accurate, as prior works such as VBMG [1], V2Meow [2], and VMAS [3] have already addressed general video-to-music generation focusing on both semantic alignment and rhythmic synchronization. Therefore, this task is not new. The authors should clearly articulate their specific contributions beyond these existing methods to justify the novelty of their work.
> >
> > (2) The authors claim, "M²UGen, from the perspective of music generation, is a strong baseline, because it incorporates a strong music generator, MusicGen."
> > However, MusicGen [4] was introduced in June 2023 ([arXiv link](https://arxiv.org/abs/2306.05284v1)), and newer, more advanced music generators such as AudioLDM2 [5], MusicLDM [6], and Stable-Audio-Open [7] have since been developed. Additionally, M²UGen is a multi-task model not specifically designed for video-to-music generation. I agree with Reviewer cqcQ's assessment that "M²UGen is a weak baseline and performs poorly on the video-to-music (V2M) task."
> >
> > (3) The authors state, "The ultimate quality of music generation is not the focus of this paper; it only needs to be satisfactory. The emphasis of this paper is on the semantic alignment and rhythmic synchronization."
> > However, audio quality is a critical aspect of music generation tasks. Even if semantic alignment is prioritized, the paper does not sufficiently explain or analyze this component, as evidenced by the unclear explanation of the SIM metric in the paper and in Response to Reviewer cqcQ (Part 1/N). The implementation details of SIM need to be clarified further.
> >
> >
> >
> >
> > References:
> >
> > [1] Zhuo L, Wang Z, Wang B, et al. Video background music generation: Dataset, method and evaluation[C]//Proceedings of the IEEE/CVF International Conference on Computer Vision. 2023: 15637-15647.
> >
> > [2] Su K, Li J Y, Huang Q, et al. V2Meow: Meowing to the Visual Beat via Video-to-Music Generation[C]//Proceedings of the AAAI Conference on Artificial Intelligence. 2024, 38(5): 4952-4960.
> >
> > [3] Lin Y B, Tian Y, Yang L, et al. VMAS: Video-to-Music Generation via Semantic Alignment in Web Music Videos[J]. arXiv preprint arXiv:2409.07450, 2024.
> >
> > [4] Copet J, Kreuk F, Gat I, et al. Simple and controllable music generation[J]. Advances in Neural Information Processing Systems, 2024, 36.
> >
> > [5] Liu H, Yuan Y, Liu X, et al. Audioldm 2: Learning holistic audio generation with self-supervised pretraining[J]. IEEE/ACM Transactions on Audio, Speech, and Language Processing, 2024.
> >
> > [6] Chen K, Wu Y, Liu H, et al. Musicldm: Enhancing novelty in text-to-music generation using beat-synchronous mixup strategies[C]//ICASSP 2024-2024 IEEE International Conference on Acoustics, Speech and Signal Processing (ICASSP). IEEE, 2024: 1206-1210.
> >
> > [7] Evans Z, Parker J D, Carr C J, et al. Stable audio open[J]. arXiv preprint arXiv:2407.14358, 2024.

---

> ### Author Response · Authors · 2024-11-21
> **Response to Reviewer yVkt (Part 2/N)**
>
> ### **Lack of Justification and Explanation**
>
> **[About visual adaptors]**
>
> > 2.1 The adaptor design section lacks a clear justification. For instance, why were these three adaptor methods chosen, instead of exploring alternative multi-modal adaptors [1, 3]? Why is CLS set as the query instead of the key-value pair?
>
> 1. **Why not use the adaptor from M$^2$UGen [1]**: To achieve semantic alignment and rhythmic synchronization, we need to compress high-frame-rate video features to fit the simple non-autoregressive generator. The adaptor in M$^2$UGen does not have a compression effect, and it only samples 32 frames uniformly from all videos. This kind of adaptors are more suitable for adapting language models.
> 2. **Why not use the adaptor from LLaMA-Adaptor [3]**: The adaptor from LLaMA-Adaptor does not actually involve compressing video features, so we still need to design the specific compression strategy ourselves. Also, LLaMA-Adaptor is designed to finetune large language models in a memory efficient way and inject potential multimodel conditions into these large language models, which requires the generator to be an autoregressive model with strong comprehension capabilities, such as LLaMA. On the one hand, finetuning our model does not exert significant memory pressure; on the other hand, this method of prompt concatenation struggles to provide direct alignment information. Our experiments show that using adapting prompt concatenation like this for conditional generation is far less effective in terms of temporal alignment than the simple method of channel-wise fusion (that is, we compress the visual feature to the same shape as the audio feature, then perform element-wise addition), because the latter provides explicit position guidance. To make a comparison, we sample the 32 frames from sliced videos uniformly to create fixed-length conditions, where the topmost 10 layers are concatenated with these 32 prompts. It is worth mentioning that for a ODE/SDE-based generator, this conditioning strategy is overly complex and far less intuitive than channel-wise fusion (or even cross-attention). Here are some preliminary experimental results, where it can be observed that the adapting prompt concatenation technique is outperformed by the channel-wise fusion.
>
> | Methods | FAD | KL | IS | FD  | BCS  | BHS | SIM |
> | :-:         | :-: | :-: | :-: | :-: | :-: | :-: | :-: |
> | LLaMA-Adaptor   | 5.35|3.81|1.47|39.03|91.36 |43.24|13.15|
> | ours            | 4.28|3.52|1.63|28.15|104.17|49.23|19.18|
>
> 3. **Why is CLS set as the query**: It is stated in Section 3.1 that we borrow the idea of semantic aware masking strategy from [6] to establish a semantic aware attention pooling strategy, where the CLS token is used as the query. Besides, if the CLS token is used as key/value, the compression operation is meaningless. Because in this case the output of the adaptor still has the same shape as the input feature.
>
> **[About metrics]**
>
> > 2.2 The paper introduces various metrics for evaluating model performance, but lacks explanations for each metric. For instance, in lines 446-449: “resulting in lower sound quality, diversity, and poorer synchronization,” it is unclear which metrics specifically measure sound quality, diversity, or synchronization.
>
> 1. FAD is defined to measure the similarity between the generation set and the source data set (using intermediate features of VGGish). A lower FAD indicates plausible generation or higher audio quality.
> 2. KL is defined to measure the distance between the distributions of two sets, measured at a paired sample level. A lower KL indicates higher audio quality. However, if the generation lacks of diversity and influences the distribution, KL might increase.
> 3. IS is computed by inception networks to measure both quality and diversity.
> 4. FD is similar to FAD but uses PANNs instead of VGGish.
> 5. BCS, BHS are used to measure rhythmic synchronization.
> 6. SIM is used to measure semantic alignment.
> 7. MOS-Q is used to measure audio quality.
> 8. MOS-A is used to measure fine-grained alignment and synchronization.
>
> **[About channel-wise fusion]**
>
> > Additionally, the statement “the channel-wise fusion of the visual conditioning still aids in synchronization” lacks experimental evidence to substantiate this claim.
>
> The evidence to substantiate that the channel-wise fusion of the visual conditioning also aids in synchronization can be found in the comment section "[About visual adaptors]"

---

> > ### Comment · Reviewer_yVkt · 2024-11-25
> >
> > The authors acknowledge the difficulty of measuring semantic alignment, stating, "Due to the lack of mature algorithms for fine-grained music emotion recognition, it is challenging to use objective metrics to measure the semantic alignment and synchronization between music and video." I agree with this statement; however, metrics such as the ImageBind Score[1], which are employed in existing works like M²UGen and VidMuse, could offer a stronger evaluation framework for assessing semantic alignment.
> >
> > [1] Girdhar R, El-Nouby A, Liu Z, et al. Imagebind: One embedding space to bind them all[C]//Proceedings of the IEEE/CVF Conference on Computer Vision and Pattern Recognition. 2023: 15180-15190.

---

> ### Author Response · Authors · 2024-11-21
> **Response to Reviewer yVkt (Part 3/N)**
>
> **[About ambiguous phrases]**
>
> > 2.3 Ambiguous phrases like “we believe” (lines 222, 483) and “might lead to” (lines 77, 483) appear multiple times in the paper. Clear support or reasoning should be provided for these assertions.
>
> We use tentative or cautious language because this is a scientific academic paper. Extreme or absolute descriptions deny any other possibilities or future discoveries, which is something we do not wish to promote. Here are some further discussions about the paragraphs reviewer yVkt points out.
>
> 1. **Line 222**. The "considers global features" part does not need further verification, because the Softmax operation guarantees that the weights at all spatial positions are positive. For the "selectively captures local information" part, we calculated the average standard deviation of the weights of each frame during the inference time: average standard deviation = 0.0131. For reference, the standard deviation of a 14x14 matrix where only one element equals 1 and all other elements equal 0 is 0.0714. Meanwhile, the expectation of standard deviation of a matrix that conforms to a normal distribution and is then processed by a Softmax function is 0.0064. Therefore, the adaptor compresses local information selectively. In addition, intuitive changes in attention can also be observed in the demo samples.
> 2. **Line 484**. This is a possible interpretation of ours for the decline in generalization ability after droping contrastive learning, as we found it tends to memorize the entire training set directly. If we compute the CLAP similarity while infering the training set, we found that the similarity of the method without contrastive learning (0.36) is much higher than that with contrastive learning (0.31).
> 3. **Line 77**. This is also a possible interpretation for the decline in generalization ability after droping certain negative samples. The quantative analysis can be found in Table 5 and Section 4.4.
>
> ### **Presentation and Writing**
>
> **[About integration of foley and sound effects]**
>
> > 3.1 The introduction (line 63) highlights tackling “Integration of foley and sound effects,” yet no further details or experiments addressing this topic are provided in the rest of the paper.
>
> This paragraph aimed to clarify the difference between the imitation of foley sounds in the generated music and the traditional approach to creating foley sounds. Many audiovisual artworks employ the technique of mimicking natural sounds with musical instruments to achieve more expressive audiovisual effects, so our method will also possess this capability after training. As mentioned in lines 63-66, these foley effects, reproduced through musical instruments, represent a special form of music generation. Many similar cases can also be found in the demo samples (for sample, at the begining of [this sample](https://muvi-v2m.github.io/data/result/video/tom_and_jerry_01[90to110](2).mp4), when Jerry hit Tom in the back of the head with a revolver, this sound is simulated by a set of percussion instruments and a brief string section). Nevertheless, the reviewer's perspective, which suggests that this does not actually fit the definition of foley sound generation and is essentially just a musical technique, also has merit. Therefore, we will refine this part to emphasize more on musical and instrumental techniques, rather than foley sound generation.

---

> > ### Comment · Reviewer_yVkt · 2024-11-25
> >
> > (1) The repeated use of phrases like "we believe" and "might lead to" undermines the strength of the paper's claims. While cautious language is necessary for academic writing to ensure rigor, a top-conference paper should base its conclusions on rigorous experiments and detailed analyses rather than speculative or tentative language. The authors need to provide stronger evidence to support their claims.
> >
> > (2) I understand the motivation behind “mimicking natural sounds with musical instruments to achieve more expressive audiovisual effects.” However, the authors claim to “aim to tackle these long-standing challenges of video-to-music generation,” explicitly including “Integration of foley and sound effects” as one of their focuses. Despite this emphasis, there is no evidence of substantial effort toward this goal in the form of experiments or analyses.

---

> ### Author Response · Authors · 2024-11-21
> **Response to Reviewer yVkt (Part 4/N)**
>
> **[About temporal shift]**
>
> > 3.2 The temporal shift method introduced in Section 3.2 is motivated as a significant contribution, but it lacks a clear explanation. Additionally, some symbols, like "m," are redefined multiple times, "C'" is not defined, which may cause confusion for readers.
>
> The symbol "m" in lines 267-268 is improperly used, as it should be used to indicate any features from the music source in other texts. Therefore, we will refine the text and change the "m" here to "b" to make a distinction. As for "C", we don't see any symbo "C" in the temporal shift section. We hope that reviewer yVkt can clarify this issue clearly.
>
> In addition, reviewer yVkt mentioned that the temporal shift section in Section 3.2 lacks a clear explanation, but did not specify which parts were difficult to understand or potentially confusing, except the improper use of symbols. Therefore, we can only provide a general explanation once again. To create a negative pair, we temporally shift the audio track to create asynchrony. Specifically, the shift offset is restricted by the minimum BPM of the music track. Because the music data we used inherently has unstable rhythm, the BPM is not constant through time. We use a dynamic beat tracking algorithm mentioned in the text to obtain a time-varying BPM sequence for each track, and select the minimum value to restrict temporal shift (if we select the maximum value, during the shift operation, there is still chance that a whole beat of lower BPM is shifted). If the minimal beat cycle is $n$ frames, we only shift $kn+b$ frames in both directions, avoiding shifting whole beats. Also, we skip the half-beat areas to avoid backbeat synchronization, so the range of $b$ is $ {⌈0.1n⌉, .., ⌊0.4n⌋, ⌈0.6n⌉, ..., ⌊0.9n⌋}$.
>
> ### **Experimental Comparison**
>
> **[Comparison with other baselines]**
>
> > 4.1 A simple baseline could have been constructed by combining an existing video understanding model with a music generation model, similar to the approach in [2, 6].
> > 4.2 The experiments omit comparisons with several relevant state-of-the-art methods, such as Diff-BGM [5], VidMuse [6], and Dance2Music-Diffusion [7].
>
> Reviewer yVkt mentioned that "a simple baseline could have been constructed by combining an existing video understanding model with a music generation model", and yes, that is why we construct MuVi(beta), an existing visual model (CLIP-ViT + attention, without contrastive pre-training) and a simple flow-matching-based generator. However, reviewer yVkt raised concerns during the "Question" section 3, stating that this baseline is simple and suboptimal and therefore unfair. We find this point to be contradictory. We will discuss more about this issue in the corresponding "Question" section.
>
> Reviewer yVkt mentioned that we omitted comparisons with other SOTA methods [2, 7, 8, 9]. Therefore, we clarify this issue and provide additional experiments as requested by reviewer yVkt here.
>
> 1. VMAS [2] has still not released their code, and they released their paper on September 11, which essentially means our work was conducted concurrently. In addition, our own replication of their method produced unsatisfactory results. Consequently, we abandoned this unfair comparison.
> 2. The situation with VidMuse [7] is similar. They only released their code on October 14th, making any comparison before that date unfair. Nevertheless, we managed to conduct a comparison with VidMuse during the rebuttal phase, using the released checkpoints. The results are shown below, where we can see that VidMuse is outperformed by our method.
> 3. Diff-BGM [8] generates symbolic music, which requires a symbolic waveform synthesizer to transform the generated music score to audio. In fact, the whole set of metrics is different between Diff-BGM and acoustic generators like ours. Therefore, we don't understand why reviewer yVkt requires us to compare with them.
> 4. Dance2Music-Diffusion [9], like other dance2music methods, aims to generate music tracks for dancing movements. It incorporates special designs for human pose and movement understanding, which is not for general videos, resulting in a unfair comparison. Nevertheless, we still made every effort to test their model and provided this comparison in response to reviewer yVkt's request. The results are listed below, where we can see that Dance2Music-Diffusion is outperformed by our method.
>
> | Methods               | FAD | KL | IS | FD  | BCS  | BHS | SIM |
> | :-:         | :-: | :-: | :-: | :-: | :-: | :-: | :-: |
> | VidMuse               | 8.13|4.88|1.50|43.82|81.35 |36.12|3.30 |
> | Dance2Music-Diffusion | 9.42|4.69|1.39|46.11|84.45 |35.91|2.39 |
> | ours                  | 4.28|3.52|1.63|28.15|104.17|49.23|19.18|

---

> > ### Comment · Reviewer_yVkt · 2024-11-25
> >
> > (1) The authors use a (CLIP-ViT + attention, without contrastive pre-training) setting as a baseline. This is not only one of the authors' own settings but also a deliberately weak configuration. Using such a baseline weakens the persuasiveness of their comparative evaluation.
> >
> > (2) While the authors state that "making any comparison before that date unfair," the paper includes only one comparative method apart from additional rebuttal experiments. This is insufficient, as many existing works in video-to-music generation, such as M²UGen[1], V2Meow[2], VidMuse[3], and VMAS[4], already provide multiple baselines for reference. Even if direct comparisons are deemed unfair due to timing, the authors could have drawn insights or evaluations from these established baselines to strengthen the validity of their proposed approach.
> >
> > [1] Liu S, Hussain A S, Sun C, et al. M $^{2} $ UGen: Multi-modal Music Understanding and Generation with the Power of Large Language Models[J]. arXiv preprint arXiv:2311.11255, 2023.
> >
> > [2] Su K, Li J Y, Huang Q, et al. V2Meow: Meowing to the Visual Beat via Video-to-Music Generation[C]//Proceedings of the AAAI Conference on Artificial Intelligence. 2024, 38(5): 4952-4960.
> >
> > [3] Tian Z, Liu Z, Yuan R, et al. Vidmuse: A simple video-to-music generation framework with long-short-term modeling[J]. arXiv preprint arXiv:2406.04321, 2024.
> >
> > [4] Lin Y B, Tian Y, Yang L, et al. VMAS: Video-to-Music Generation via Semantic Alignment in Web Music Videos[J]. arXiv preprint arXiv:2409.07450, 2024.

---

> ### Author Response · Authors · 2024-11-21
> **Response to Reviewer yVkt (Part 5/N)**
>
> **[About finetuing M2Ugen]**
>
> > 4.3 The M^2Ugen method shows comparable or superior results in terms of audio quality (Table 2). Fine-tuning this method on the dataset used in this paper could provide additional insight into its performance.
>
> It is worth mentioning that the music-visual alignment and the sound quality are somewhat of a trade-off. A similar phenomenon can be seen in Figure 4 that higher CFG scale enhances the alignment while diminishing sound quality in terms of IS. Nevertheless, we managed to finetune the M$^2$UGen by the several steps described in their original paper: 1) we slice the videos beforehand, and use MU-LLaMA to generate captions for audios; 2) we use MPT-7B to generate answers for training the LLaMA 2 model; 3) we finetune the LLaMA 2 model using the LoRA and the adaptor technique, where we have to crop the video frames to fit the ViViT encoder; 4) we finetune the MusicGen medium model with the generated captions until convergence. Preliminary experimental results are listed below, from which it can be observed that the performance of M$^2$UGen did not change significantly after finetuning, and even slightly decreased. This is because their design lacks considerations for audio-visual temporal alignment, and the finetuing on a smaller dataset may cause possible overfitting in large models. In fact, using language models as a bridge results in the loss of a substantial amount of effective visual information.
>
> | Methods                 | FAD | KL | IS | FD  | BCS  | BHS | SIM |
> | :-:         | :-: | :-: | :-: | :-: | :-: | :-: | :-: |
> | M$^2$UGen               | 5.12|3.83|1.65|32.14|75.21 |25.14|1.41 |
> | M$^2$UGen (finetuned)   | 5.23|3.96|1.57|33.82|75.42 |25.05|1.38|
> | ours                    | 4.28|3.52|1.63|28.15|104.17|49.23|19.18|
>
> ### **Questions**
>
> **[About local and global features]**
>
> > 1.Claim on Previous V2M Methods (lines 39-41): The authors claim that "Previous V2M methods focus on global features," presenting this as a limitation of past approaches. However, this appears inconsistent with prior work, as several existing methods focus on local clip features for training. For instance, V2Meow [1] and VMAS [2] emphasize local clip-based features, while VidMuse [3] captures both local and global features through long-short-term modeling. The authors should clarify and provide evidence to support their assertion about the emphasis on global features in previous V2M approaches.
>
> We agree with reviewer yVkt that the statement "previous V2M methods focus on global features" is inaccurate. The mentioned methods [2, 7, 10] indeed incorporate frame-level visual representations. However, VidMuse [7] mainly focuses on semantic alignment, V2Meow [10] mainly focuses on rhythmic synchronization. In fact, V2Meow does incorporate frame-level semantic features (CLIP) to generate music, but their evaluation mainly focuses on global semantic relevance, not fine-grained semantic alignment. As for VMAS [2], they released their paper on September 11, which essentially means our work was conducted concurrently. Also, from their demo samples, it can be observed that the music does not undergo observable emotional changes with the video; most videos maintain a single emotional style, and there is no clear rhythmic synchronization.
>
> **[About beat synchronization metrics]**
>
> > 2.Choice of Beat Synchronization Metrics and Exclusion of Dance Video Music Generation for Comparison: The authors select Beats Coverage Score (BCS) and Beats Hit Score (BHS) as metrics to evaluate beat synchronization, following the approach in [4] (line 346), which specifically targets music generation for dance videos. However, the authors then claim in line 364 that "D2M-GAN are not considered for comparison because their scope of application differs from ours." If dance-related videos are outside MuVi’s intended scope, it is unclear why dance-specific metrics are being applied for evaluation. This raises a need for clarification.
>
> We disagree with reviewer yVkt's opinion that BCS and BHS are dance-specific metrics, and that using these metrics implies comparing with dance2music methods. We believe that as technology advances, people will gradually uncover the essence of problems, rather than just focusing on their surface. The main focus of our work differs from that of dance2music methods. In fact, the model inputs are majorly different, as they require specific human motion encoding or body keypoints. However, it must be acknowledged that these two tasks share similarities in evaluation, as they both focus on the rhythmic synchronization of the generated music. Therefore, rather than saying this metric is dance-specific, it is more accurate to say it is rhythm-specific, and we borrow it to measure rhythmic synchronization.

---

> > ### Comment · Reviewer_yVkt · 2024-11-25
> >
> > The fine-tuned M²UGen model performs worse on key metrics like FAD, KL, and FD, which is counterintuitive.
> > The authors' explanation for this performance drop (e.g., overfitting or lack of temporal alignment) is unconvincing. A deeper analysis is needed to justify these results.

---

> ### Author Response · Authors · 2024-11-21
> **Response to Reviewer yVkt (Part 6/N)**
>
> **[About MuVi(beta)]**
>
> > 3.Choice of MuVi(beta) Setting for Comparison: The paper claims "use CLIP-ViT(base-patch16) and the attention pooling adaptor as the visual encoder" for MuVi(beta) (lines 366-367). However, Table 1 shows that the VideoMAE V2 with a Softmax adaptor yields better results for this setting. It is unclear why a suboptimal setting was selected for MuVi(beta), as this choice could impact the fairness and interpretability of the comparisons. An explanation from the authors on the rationale for this choice would provide more clarity.
>
> As mentioned in the "[Comparison with other baselines]" section, we construct MuVi(beta) to create a simple and trivial baseline, as requested in the Weakness 4.1 of reviewer yVkt's comments, since we don't have much baselines to compare with. MuVi(beta) is not only constructed with CLIP + attention pooling, it is also trained without contrastive pre-training, resulting in a very simple baseline. Reviewer yVkt accuses us of choosing this model for comparison because it performs poorly, which is a case of putting the cart before the horse. We choose CLIP + attention combination as the adaptor strategy, because it is actually very competitive comparing with the VideoMAE V2 + Softmax combination. Also, using CLIP as the visual encoder is a common choice among other works, meeting the requirements of a simple baseline.
>
> ---
>
> We hope our clarifications address your concerns and we are looking forward to your re-assessment of our work. We also welcome further discussion with you. Thank you again for your efforts.
>
> **References**
>
> [1] Liu S, Hussain A S, Sun C, et al. M2UGen: Multi-modal Music Understanding and Generation with the Power of Large Language Models[J]. arXiv preprint arXiv:2311.11255, 2023.
>
> [2] Lin Y B, Tian Y, Yang L, et al. VMAS: Video-to-Music Generation via Semantic Alignment in Web Music Videos[J]. arXiv preprint arXiv:2409.07450, 2024.
>
> [3] Zhang R, Han J, Liu C, et al. Llama-adapter: Efficient fine-tuning of language models with zero-init attention[J]. arXiv preprint arXiv:2303.16199, 2023.
>
> [4] Radford A, Kim J W, Hallacy C, et al. Learning transferable visual models from natural language supervision[C]//International conference on machine learning. PMLR, 2021: 8748-8763.
>
> [5] Rombach R, Blattmann A, Lorenz D, et al. High-resolution image synthesis with latent diffusion models[C]//Proceedings of the IEEE/CVF conference on computer vision and pattern recognition. 2022: 10684-10695.
>
> [6] Hou Z, Sun F, Chen Y K, et al. Milan: Masked image pretraining on language assisted representation[J]. arXiv preprint arXiv:2208.06049, 2022.
>
> [7] Tian Z, Liu Z, Yuan R, et al. VidMuse: A Simple Video-to-Music Generation Framework with Long-Short-Term Modeling[J]. arXiv preprint arXiv:2406.04321, 2024.
>
> [8] Li S, Qin Y, Zheng M, et al. Diff-BGM: A Diffusion Model for Video Background Music Generation[C]//Proceedings of the IEEE/CVF Conference on Computer Vision and Pattern Recognition. 2024: 27348-27357.
>
> [9] Zhang C, Hua Y. Dance2Music-Diffusion: leveraging latent diffusion models for music generation from dance videos[J]. EURASIP Journal on Audio, Speech, and Music Processing, 2024, 2024(1): 48.
>
> [10] Su K, Li J Y, Huang Q, et al. V2Meow: Meowing to the Visual Beat via Video-to-Music Generation[C]//Proceedings of the AAAI Conference on Artificial Intelligence. 2024, 38(5): 4952-4960.

---

> ### Author Response · Authors · 2024-11-24
> **Looking Forward to Further Feedback**
>
> Dear Reviewer yVkt,
>
> Thank you again for your great efforts and valuable comments.
>
> We have tried to address the main concerns you raised in the review and made huge efforts such as additional experiments. As the end of the rebuttal phase is approaching, we are looking forward to hearing your feedback regarding our answers. We are always happy to have a further discussion and answer more questions (if any).
>
> Thanks in advance,
>
> Submission10758 Authors

---

> > ### Comment · Reviewer_yVkt · 2024-11-25
> > **Additional Concerns**
> >
> > I agree with Reviewer JLkN’s remark that "the collected video mostly includes Disney, Tom and Jerry, and Silent Films. I think this fact should also be described in the paper well since the proposed method is valid only on these kind of video contents for now."
> >
> > Moreover, there are potential concerns about data leakage, raising doubts about the model’s generalizability and its tendency to overfit on the authors’ dataset:
> >
> > (1) On the demo page, the generated music in almost all samples predominantly features strings, brass instruments, piano, and percussion. This closely matches the training data categories mentioned by the authors and does not reflect the diversity of the MTG-Jamendo Dataset, which includes 95 genres and 41 instruments. The authors also claim that the MTG-Jamendo Dataset is used to train the music generator, but the lack of diversity in the generated music suggests possible overfitting to specific subsets of the training data.
> >
> > (2) For video samples from categories similar to those in the training dataset, such as [sample_1](https://muvi-v2m.github.io/data/result/video/tom_and_jerry_01%5B90to110%5D(2).mp4), [sample_2](https://muvi-v2m.github.io/data/result/video/tom_and_jerry_01[270to290](0).mp4), the model demonstrates excellent rhythmic synchronization. However, for videos from other categories, such as [sample_3](https://muvi-v2m.github.io/data/result/video/game_cg[80to100](6).mp4), [sample_4](https://muvi-v2m.github.io/data/result/video/game_cg[540to560](1).mp4), the performance deteriorates significantly, particularly in terms of synchronization and musical alignment.
> >
> > The authors claim that "we believe this method has a certain degree of general applicability." However, this claim is not adequately supported. To validate this assertion, evaluating the model's performance on established and diverse benchmarks, such as LORIS [1], SymMV [2], V2M [3], AIST++ [4], or BGM909 [5], would help comprehensively assess its generalizability and robustness across different types of content.
> >
> >
> > [1] Yu J, Wang Y, Chen X, et al. Long-term rhythmic video soundtracker[C]//International Conference on Machine Learning. PMLR, 2023: 40339-40353.
> >
> > [2] Zhuo L, Wang Z, Wang B, et al. Video background music generation: Dataset, method and evaluation[C]//Proceedings of the IEEE/CVF International Conference on Computer Vision. 2023: 15637-15647.
> >
> > [3] Tian Z, Liu Z, Yuan R, et al. VidMuse: A simple video-to-music generation framework with long-short-term modeling[J]. arXiv preprint arXiv:2406.04321, 2024.
> >
> > [4] Li R, Yang S, Ross D A, et al. AI choreographer: Music conditioned 3D dance generation with AIST++[C]//Proceedings of the IEEE/CVF International Conference on Computer Vision. 2021: 13401-13412.
> >
> > [5] Li S, Qin Y, Zheng M, et al. Diff-BGM: A Diffusion Model for Video Background Music Generation[C]//Proceedings of the IEEE/CVF Conference on Computer Vision and Pattern Recognition. 2024: 27348-27357.

---

> ### Author Response · Authors · 2024-11-25
> **Replying to Official Comment by Reviewer yVkt (1/2)**
>
> **[About novelty and contribution]**
>
> 1. Reviewer yVkt mentions three works, VBMG [1], V2Meow [2], and VMAS [3], that "have already addressed general video-to-music generation focusing on both semantic alignment and rhythmic synchronization". It is worth emphasizing that the "alignment" in our context is interpreted as local or fine-grained alignment (lines 39-41). That is, if the mood or the rhythm of the scene changes, the musical features must respond immediately. However, VBMG [1] only generates music with a constant rhythm, and both VBMG and V2Meow [2] only focus on global semantic alignment (they incorporate fine-grained semantic features as input, but only evaluate regarding global contents). VMAS [3] is a concurrent work, so we do not categorize it as 'previous' work.
> 2. M$^2$UGen, from the perspective of music generation, is a strong baseline, because the music generator it involves, MusicGen, has ten times our parameter count and four times our training data volume.
> 3. While achieving fine-grained semantic alignment rhythmic synchronization, the proposed method reaches a level of sound quality that is competitive with methods incorporating text-to-music generation models. The explanation of the SIM metric is discussed in the following comment section.
>
> **[About the SIM metric]**
>
> The SIM metric is a reference-free metric derived from the contrastively pre-trained encoders. Specifically, the SIM value is the average of frame-level cosine similarity values, which can handle varying-length music-visual pairs. For the music track, an AudioMAE encoder encodes the audio in a fine-grained way and a learnable linear layer transforms the encoding into an audio feature that has the same length as the compressed video feature. The implementation details of the encoders are listed in Appendix C. Given the audio and visual features both in the shape of (N, C), we compute the element-wise cosine similarity values and compute the average to be the SIM value. Therefore, the SIM value pays attention to local similarity.
>
> **[About integration of foley and sound effects]**
>
> The intent of this paragraph was to distinguish the mimicking of foley sounds in the generated music from the logic of traditional foley sound generation. This raises a very important issue: the objects in the video are not necessarily the sources of the sounds being generated. This renders some traditional video-sound or video-music datasets ineffective, as they are constructed based on the pairing of the sound-producing object and the sound produced. This also explains why we do not use traditional datasets or models/metrics developed based on these datasets (such as ImageBind). We will revise the manuscript to make this section more precise.
>
> **[About MuVi(beta)] and baseline comparison**
>
> 1. We create a simple and intuitive baseline with the (CLIP-ViT + attention, without contrastive pre-training) setting, which is also supported by reviewer yVkt ("A simple baseline could have been constructed by combining an existing video understanding model with a music generation model, similar to the approach in [VMAS, VidMuse]").
> 2. As for the two baselines reviewer yVkt mentioned: VMAS is a concurrent work and has not released their code; VidMuse has not released their code until October 14th. We have made every effort to conduct additional experiments and listed the results in the original rebuttal comments (the "[Comparison with other baselines]" section). We list the results here again:
>
> |Methods|FAD|KL|IS|FD|BCS  | BHS | SIM |
> |:-: | :-: | :-: | :-: | :-: | :-: | :-: | :-: |
> |VidMuse|8.13|4.88|1.50|43.82|81.35 |36.12|3.30 |
> |ours|4.28|3.52|1.63|28.15|104.17|49.23|19.18|
>
> 3. Reviewer yVkt mentions four existing works in video-to-music generation, M$2$UGen [4], V2Meow [2], VidMuse [5], and VMAS [3], and the paper "includes only one comparative method apart from additional rebuttal experiments". However, V2Meow has not released their code, and even their [demo page](https://tinyurl.com/v2meow) is inaccessible. VidMuse has not released their code until October 14th, and our initial replication also produced unsatisfactory results. VMAS is essentially a concurrent work, because they released the paper on September 11 UTC.
>
> **References**:
>
> [1] Zhuo L, Wang Z, Wang B, et al. Video background music generation: Dataset, method and evaluation[C]//Proceedings of the IEEE/CVF International Conference on Computer Vision. 2023: 15637-15647.
>
> [2] Su K, Li J Y, Huang Q, et al. V2Meow: Meowing to the Visual Beat via Video-to-Music Generation[C]//Proceedings of the AAAI Conference on Artificial Intelligence. 2024, 38(5): 4952-4960.
>
> [3] Lin Y B, Tian Y, Yang L, et al. VMAS: Video-to-Music Generation via Semantic Alignment in Web Music Videos[J]. arXiv preprint arXiv:2409.07450, 2024.
>
> [4] Liu S, Hussain A S, Sun C, et al. M$^2$UGen: Multi-modal Music Understanding and Generation with the Power of Large Language Models[J]. arXiv preprint arXiv:2311.11255, 2023.

---

> > ### Comment · Reviewer_yVkt · 2024-11-25
> >
> > [About M²UGen]
> >
> > The criterion for determining whether a model is a strong music generator should not be based on model size but rather on the quality of the generated music. According to the authors' logic, does this imply that MusicLM, the early text-to-music generation model with 430M parameters, is stronger than the music generator proposed by the authors (with 330M parameters)? Or that the AudioLDM-2 Full size with 346M parameters is weaker than MusicGen?
> >
> >
> > [About the Baselines]
> >
> > M²UGen compares its video-to-music generation task with two baselines (CoDi and CMT), while V2Meow compares three methods (CDCD, D2M-GAN, and CMT) in its paper and reports scores on the AIST++ benchmark. VidMuse and VMAS, on the other hand, compared with five additional methods in their respective papers.

---

> ### Author Response · Authors · 2024-11-25
> **Replying to Official Comment by Reviewer yVkt (2/2)**
>
> **[About finetuning M$^2$Ugen]**
>
> The generation process of M$^2$Ugen is dissected: the LLaMA 2 decoder summarizes the main theme of the visual features extracted by the ViViT model and generates a description about the feature of the music to generate. The MusicGen generator then follows the instruction to generate music. Therefore, in our task, what actually has a direct impact on the generation quality is the finetuning step of MusicGen. Specifically, if we only finetune the LLaMA 2 model and keep the MusicGen model unchanged, the objective results essentially remain still in multiple runs' average. In addition, FAD, KL, and IS also indicate the latent distribution distance between the generation set and the source set. A worse value also implies that the generated samples have relatively little similarity to the dataset.
>
> **[About generalizability]**
>
> The main focus of this paper is to explore the way to generate music tracks with fine-grained semantic alignment and rhythmic synchronization, and exploring its generalizability is not the primary focus. Even so, we will make every effort to evaluate the performance on LORIS [6] benchmark. The reason we choose LORIS but the others are: SymMV [7] is for symbolic music generation; V2M [5] has not released their test dataset; AIST++ [8] is included in the LORIS benchmark. However, as the rebuttal deadline approaches, if we are unable to complete this evaluation before the deadline, please understand.
>
> **References**:
>
> [1] Zhuo L, Wang Z, Wang B, et al. Video background music generation: Dataset, method and evaluation[C]//Proceedings of the IEEE/CVF International Conference on Computer Vision. 2023: 15637-15647.
>
> [2] Su K, Li J Y, Huang Q, et al. V2Meow: Meowing to the Visual Beat via Video-to-Music Generation[C]//Proceedings of the AAAI Conference on Artificial Intelligence. 2024, 38(5): 4952-4960.
>
> [3] Lin Y B, Tian Y, Yang L, et al. VMAS: Video-to-Music Generation via Semantic Alignment in Web Music Videos[J]. arXiv preprint arXiv:2409.07450, 2024.
>
> [4] Liu S, Hussain A S, Sun C, et al. M$^2$UGen: Multi-modal Music Understanding and Generation with the Power of Large Language Models[J]. arXiv preprint arXiv:2311.11255, 2023.
>
> [5] Tian Z, Liu Z, Yuan R, et al. Vidmuse: A simple video-to-music generation framework with long-short-term modeling[J]. arXiv preprint arXiv:2406.04321, 2024.
>
> [6] Yu J, Wang Y, Chen X, et al. Long-term rhythmic video soundtracker[C]//International Conference on Machine Learning. PMLR, 2023: 40339-40353.
>
> [7] Zhuo L, Wang Z, Wang B, et al. Video background music generation: Dataset, method and evaluation[C]//Proceedings of the IEEE/CVF International Conference on Computer Vision. 2023: 15637-15647.
>
> [8] Li R, Yang S, Ross D A, et al. AI choreographer: Music conditioned 3D dance generation with AIST++[C]//Proceedings of the IEEE/CVF International Conference on Computer Vision. 2021: 13401-13412.

---

> > ### Comment · Reviewer_yVkt · 2024-11-25
> >
> > [About generalizability]
> >
> > The authors now claim that "exploring its generalizability is not the primary focus." However, earlier in response to Reviewer JLkN’s comment, they stated, "we believe this method has a certain degree of general applicability." If the authors still wish to claim this, the claim needs to be substantiated. Additionally, the concern regarding potential data leakage should be addressed and clarified, with support from the authors to resolve these issues.
> >
> > [About the benchmark]
> >
> > V2M has not been released, but both SymMV and BGM909, despite being proposed for symbolic music generation tasks, contain audio music modalities and corresponding ground truth, which can be used for evaluation. The authors could choose one of these for comparison, or alternatively, they could select AIST++ or LORIS.
> >
> > Another point worth mentioning is that all materials should have been completed and submitted by the submission deadline. However, as I previously mentioned to the Area Chair, the authors continued to modify their GitHub repository after the submission deadline, even during the review period, as evidenced by https://github.com/MuVi-V2M/MuVi-V2m.github.io/commits/main. This could be viewed as unfair and provides the authors with an advantage.
> >
> > That being said, if the Area Chair agrees, I suggest that the authors include the new results, including new baselines and new benchmarks, as references. If the authors are unable to provide these results before the rebuttal deadline, I recommend they run the demo pages for baselines like V2Meow, VidMuse, and VMAs and present these results as a reference instead.

---

> ### Author Response · Authors · 2024-12-02
> **Replying to Official Comment by Reviewer yVkt for Additional Experimental Results (1/2)**
>
> We apologize for the delayed response, as replicating and comparing baseline methods takes a considerable amount of time, especially since some baseline methods require a significant amount of time for inference (e.g., VidMuse needs 14 min to generate a 25s audio clip, while MuVi only needs 6s). We still have some experiments that are not completed, but we will do our best to finish them and provide partial results first. Please understand. We will also include the additional results in the revised manuscript.
>
> **[ImageBind Evaluation]**
>
> We have done the ImageBind AV score evaluation as requested by reviewer yVkt, and the results are listed below.
>
> | Methods               | ImageBind AV score |
> | :-:         | :-: |
> | VidMuse               | 0.0527|
> | M2UGen                | 0.0513|
> | ours                  | 0.0542|
>
> It is worth mentioning that, because the audio-visual bind of ImageBind is trained with Audioset dataset, which is constructed based on the pairing of the sound-producing object and the sound produced (that is, traditional video-to-audio dataset), the ImageBind AV score is invalid and irrelevant for our task (we have already elaborated this in the "Integration of foley and sound effects" paragraph in the original paper). Although Audioset contains over 1M music samples, most of them are performance videos, i.e., videos that record the sound of certain musical instruments. This essentially categorizes them as video-to-audio type data.
>
> Our task does not require such stringent relationships, a video of a car does not imply the sound of its engine, nor does a video of a violin necessarily imply the sound of the violin. Therefore, using this metric for our task is unreasonable. Just because other works (VidMuse and M$^2$UGen, mentioned by reviewer yVkt) have used this metric does not mean it is reasonable. **This was our initial reason for not using this metric.** However, we have included the results here, which demonstrate that our method still outperforms the baselines.

---

> ### Author Response · Authors · 2024-12-02
> **Replying to Official Comment by Reviewer yVkt for Additional Experimental Results (2/2)**
>
> **[Additional Benchmark]**
>
> Evaluation results on 25s dancing test set (AIST++) from LORIS benchmark:
>
> | Methods               | FAD | KL  | IS  | FD   | BCS   | BHS  | SIM  | BCS' | CSD  | BHS' | HSD  | F1   | MOS-Q | MOS-A | ImageBind AV score |
> | :-:                   | :-: | :-: | :-: | :-: | :-: | :-: | :-: | :-: | :-: | :-:  | :-: | :-: | :-: | :-: | :-: |
> | GT                    | -   | -   | -   | -    | -     | -    | -    | -    | -    | -    | -    | -    | -      | -      | 0.0527                  |
> | LORIS                 | -   | -   | -   | -    | -     | -    | -    | 98.6 | 6.1  | 90.8 | 13.9 | 94.5 | -      | -      | -                  |
> | CMT                   | 8.19| 4.81| 1.12| 73.26| 90.85 | 39.38| 3.25 | 96.9 | 6.3  | 46.0 | 18.4 | 62.4 | 3.36   | 3.41   | 0.0508                  |
> | VidMuse               | 5.63| 2.12| 1.19| 34.01| 90.07 | 41.14| 5.12 | 95.7 | 12.8 | 97.0 | 6.6  | 96.4 | 3.58   | 3.54   | 0.0601                  |
> | M2UGen                | 5.80| 4.56| 1.45| 40.42| 65.76 | 24.44| 3.85 | 94.6 | 3.7  | 91.3 | 17.5 | 93.6 | 3.81   | 2.86   | 0.0552                  |
> | ours                  | 5.56| 2.06| 1.77| 33.78| 125.72| 46.23| 14.25| 95.9 | 13.0 | 59.5 | 26.4 | 73.4 | 3.85   | 4.03   | 0.0513                  |
>
> Since the evaluation is based on LORIS benchmark, we also borrow the corresponding metrics (BCS', CSD, BHS', HSD, F1) to make a fair comparison. Because the checkpoints of LORIS are inaccessible, we only copy the results from their orirginal paper. It is worth mentioning that the metrics from LORIS are computed second-wise, while ours (BCS, BHS) are computed based on a 100ms tolerance. **This makes LORIS metrics significantly loose for rhythmic synchronization, namely, models can perform very well or even cheat on these metrics easily. This was also our initial reason for not using this metric.**
>
> It is worth mentioning that our model has never seen any specially collected dancing videos, not to mention the whole training split of LORIS dataset. Therefore, our model is tested in an out-of-domain fashion.
>
> Another important thing is that the ImageBind AV score is pretty low even for ground truth (GT) samples. This just confirms our previous point that this metric is inappropriate in this task. To address this issue, we conducted additional subjective evaluations and also introduced metrics specifically for semantic alignment and rhythmic synchronization. The results demonstrate the superior performance of the proposed method.
>
> Evaluation results on 25s figure skating test set from LORIS benchmark:
>
> | Methods               | FAD | KL  | IS  | FD   | BCS   | BHS  | SIM  | BCS' | CSD  | BHS' | HSD  | F1   | MOS-Q | MOS-A | ImageBind AV score |
> | :-:                   | :-: | :-: | :-: | :-: | :-: | :-: | :-: | :-: | :-: | :-:  | :-: | :-: | :-: | :-: | :-: |
> | GT                    | -   | -   | -   | -    | -     | -    | -    | -    | -    | -    | -    | -    | -      | -      | 0.0578                  |
> | LORIS                 | -   | -   | -   | -    | -     | -    | -    | 52.2 | 18.5 | 57.0 | 19.8 | 54.5 | -      | -      | -                  |
> | CMT                   |16.76| 3.16| 1.19| 83.24| 82.03 | 38.64| 3.59 | 39.3 | 28.5 | 75.1 | 27.6 | 51.6 | 3.25   | 3.36   | 0.0521                  |
> | VidMuse               |12.62| 2.85| 1.29| 71.13| 80.92 | 39.84| 4.03 | 53.6 | 22.5 | 91.5 | 14.6 | 67.6 | 3.52   | 3.45   | 0.0487                  |
> | M2UGen                |14.12| 2.77| 1.25| 69.90| 82.62 | 39.12| 3.52 | 65.3 | 17.6 | 95.9 | 9.4  | 78.4 | 3.77   | 3.12   | 0.0616                  |
> | ours                  |14.44| 2.58| 1.44| 64.02| 102.60| 50.67| 15.18| 65.7 | 25.8 | 64.9 | 26.8 | 60.0 | 3.79   | 3.93   | 0.0619                  |
>
> It is worth noting that **the audio quality of ground-truth tracks of the figure skating dataset is extremely poor**, compared to other musical datasets. The audio in these videos is very noisy, mixed with a lot of noise and even human voices. Moreover, these videos do not exhibit any obvious semantic transitions. Therefore, **we believe this benchmark cannot demonstrate the real performance of our method**, but we still conducted the evaluation as requested by reviewer yVkt. From the results, we can observe a very clear variance, and the high FAD indicates that the distribution of audio features learned by the models are quite different from that of this dataset. This indirectly suggests that the dataset is lacking in terms of audio quality.

---

### Official Review · Reviewer_cqcQ · 2024-11-04

**Soundness:** 3
**Presentation:** 3
**Contribution:** 3
**Rating:** 8
**Confidence:** 3

**Summary:**

This paper proposes a novel video-to-music generation method that improves semantic similarity and rhythmic consistency by enhancing the visual encoder and adopting a new contrastive learning pretraining approach. The authors also conducted a series of ablation studies to examine the specific impact of different modules on the model’s performance.

**Strengths:**

The writing in this paper is clear, and the motivation aligns well with intuition. Overall, the experiments are reasonably thorough and adequately conducted.

The assumptions about video-to-music generation are sensible. The authors address key challenges of the task, such as temporal semantic consistency in videos, rhythm alignment, and distinguishing between sound effects and music. Therefore, the proposed improvements seem well-suited for tackling these issues.

The evaluation metrics are also well-chosen, covering metrics that sufficiently measure the model’s performance across various aspects. The experimental results are convincing.

**Weaknesses:**

The paper includes an additional section on the in-context learning (ICL) capability of music generation models, which I feel deviates from the main theme of the proposed task. From the limited experiments presented, this section does not add to the paper’s persuasiveness, nor does it convincingly demonstrate actual ICL capabilities. I suggest removing this contribution from the paper.

The discussion on semantic synchronisation is lacking. The paper initially introduces an example: “the style, melody, and emotions of the music will evolve in harmony with the video content,” and Figure 2 also touches on this aspect. The authors attempt to measure semantic synchronisation using the SIM metric and imply that Section 4.2 will discuss this metric in detail. However, I found no mention of this metric in Section 4.2. Thus, I believe the discussion on semantic synchronisation is relatively insufficient. I recommend further analysis of the model’s performance in semantic synchronisation, perhaps by including additional case studies.

Training data concerns for the music generation model. I am curious about one particular issue: since the Jamendo dataset does not include semantic variations, requiring it to generate music that aligns with video semantic changes is effectively an out-of-distribution (OOD) problem. How do the authors plan to address this point?

Choice of baseline. I fully understand the difficulty in comparing to baseline models due to the lack of open-source availability. In this situation, using M2UGen as a baseline is acceptable. However, I would like to note that M2UGen is a weak baseline and performs poorly on the video-to-music (V2M) task. While not a strict requirement, I noticed that VidMuse has released its code. Considering that this baseline is also prominently discussed in the paper, I encourage the authors to include a comparison with it.

**Questions:**

N/A

---

> ### Author Response · Authors · 2024-11-21
> **Response to Reviewer cqcQ (Part 1/N)**
>
> We thank the reviewer for the constructive and professional review and we are sorry about the unsupported claims.
>
> **[About in-context learning]**
>
> > The paper includes an additional section on the in-context learning (ICL) capability of music generation models, which I feel deviates from the main theme of the proposed task. From the limited experiments presented, this section does not add to the paper’s persuasiveness, nor does it convincingly demonstrate actual ICL capabilities. I suggest removing this contribution from the paper.
>
> The reason for including this paragraph is to demonstrate the potential controllability of the method. Without incorporating specific training strategies for ICL, our method can only generate music with random instruments and types, although the rhythm and mood of the music are aligned with the video. We believe that the controllability of music generation is necessary but not primary, hence we included only this paragraph for explanation. Also, this section highlights the capability to generate music in any style, demonstrating the generalizability on the generation side. More intuitive effects of ICL capabilities can be seen on the demo page.
>
> **[About semantic synchronisation]**
>
> > The discussion on semantic synchronization is lacking. The paper initially introduces an example: “the style, melody, and emotions of the music will evolve in harmony with the video content,” and Figure 2 also touches on this aspect. The authors attempt to measure semantic synchronisation using the SIM metric and imply that Section 4.2 will discuss this metric in detail. However, I found no mention of this metric in Section 4.2. Thus, I believe the discussion on semantic synchronisation is relatively insufficient. I recommend further analysis of the model’s performance in semantic synchronisation, perhaps by including additional case studies.
>
> Thank you for raising this issue. Although we agree with reviewer cqcQ’s point that the discussion on semantic synchronization is not sufficiently thorough, we need to first clarify some statements in the paper and address any potential misunderstandings. In line 353, the statement "which is discussed in detail in Section 4.2" was originally used to refer to the choice of using VideoMAE V2 + Softmax combination as the visual encoder in the SIM measure, not the SIM measure itself. We believe it is necessary to specify what exactly the visual encoder in this SIM measure is. Directly stating our choice of the encoder in this paragraph might seem abrupt, so we have reserved the justification for this choice for Section 4.2, where we demonstrate that this choice is superior in the final generation results, in the first paragraph of Section 4.2. This is our intention, but indeed, the phrasing can be misleading. We will revise this section accordingly.
>
> Nevertheless, the discussion on semantic synchronization is indeed relatively insufficient, although we indeed mentioned the SIM metric in line 430 to show the poor performance of M$^2$UGen in terms of synchronization. Due to the lack of mature algorithms for fine-grained music emotion recognition, it is challenging to use objective metrics to measure the semantic alignment and synchronization between music and video, especially for emotion and content transitions. As far as we know, SIM is the only objective metric currently available to us. However, during the subjective evaluation phase, we required the raters to focus more on fine-grained alignment (such as emotion transitions) in the generated music, and the results indicate a superior performance of our model. For an intuitive impression, the demo samples also demonstrate fine-grained and rapid-response emotion and content transition.

---

> ### Author Response · Authors · 2024-11-21
> **Response to Reviewer cqcQ (Part 2/N)**
>
> **[About music training data]**
>
> > Training data concerns for the music generation model. I am curious about one particular issue: since the Jamendo dataset does not include semantic variations, requiring it to generate music that aligns with video semantic changes is effectively an out-of-distribution (OOD) problem. How do the authors plan to address this point?
>
> The Jamendo dataset is not used to train the model to generate music that aligns with video. Basically, we utilize two sets of training data: the audio-only music data, and the audio-visual paired data. The audio-only data is used to pre-train the unconditional DiT to acquire a generator that has a certain general capability to generate music, or you can call it a parameter warm-up. The audio-visual paired data is used to finetune the DiT combined with the visual adaptor to empower the model to generate semantically and rhythmically aligned music with visual conditions. The reasons of pre-training are two folds: 1) we want to show that our method can be generalized to any simple music generator, because the generator is not the main focus of this paper; 2) without pre-training, this model, having been exposed to very limited data (about 280 hours of music), would greatly restrict its generalization ability and the diversity of the generated samples. Specifically, if we cancel the pre-training procedure, the performance drops dramatically, as shown in the first row of Table 5.
>
> **[About baselines]**
>
> > Choice of baseline. I fully understand the difficulty in comparing to baseline models due to the lack of open-source availability. In this situation, using M2UGen as a baseline is acceptable. However, I would like to note that M2UGen is a weak baseline and performs poorly on the video-to-music (V2M) task. While not a strict requirement, I noticed that VidMuse has released its code. Considering that this baseline is also prominently discussed in the paper, I encourage the authors to include a comparison with it.
>
> Thank you for your suggestions. VidMuse released its code on October 14th, which was after the paper submission deadline, and our own implementation of VidMuse turned out to perform much worse than our method and M$^2$UGen. Hence we did not include their results in our paper. Nevertheless, during the rebuttal phase, we managed to compare with VidMuse based on the newly released checkpoints, and the results are listed below. From the results, it can be seen that VidMuse is outperformed by our model in both terms of audio quality and synchronization. Subjectively, the music VidMuse generates contains noise and artifacts, and the rhythm is not synchronized with the video.
>
> | Methods               | FAD | KL | IS | FD  | BCS  | BHS | SIM |
> | :-:         | :-: | :-: | :-: | :-: | :-: | :-: | :-: |
> | VidMuse               | 8.13|4.88|1.50|43.82|81.35 |36.12|3.30 |
> | ours                  | 4.28|3.52|1.63|28.15|104.17|49.23|19.18|
>
> ---
>
> Once again, thank you for your effort in reviewing our work and your valuable comments, which are of great significance to improving our work.

---

> ### Author Response · Authors · 2024-11-25
> **Looking Forward to Further Feedback**
>
> Dear Reviewer cqcQ,
>
> Thank you again for your great efforts and valuable comments.
>
> We have tried to address the main concerns you raised in the review and made huge efforts such as additional experiments. As the end of the rebuttal phase is approaching, we are looking forward to hearing your feedback regarding our answers. We are always happy to have a further discussion and answer more questions (if any).
>
> Thanks in advance,
>
> Submission10758 Authors

---

> > ### Comment · Reviewer_cqcQ · 2024-11-25
> > **I appreciate your efforts improving the paper quality.**
> >
> > Thanks for your rebuttal.
> >
> > I agree that your rebuttal solved most issues I raised. Assuming you will integrate these revisions to the next version of your paper, I am happy to raise the score to 8.
> >
> > However, I keep my opinion about in-context learning section. After reading the rebuttal I understand the reason why this paper includes it, but it still kind of over-claiming. My suggestion is that you can discuss it, but do not count it as the contribution of this paper.
> >
> > Again, good rebuttal. Thanks for your work. Good luck.

---

### Official Review · Reviewer_JLkN · 2024-11-04

**Soundness:** 3
**Presentation:** 3
**Contribution:** 2
**Rating:** 6
**Confidence:** 3

**Summary:**

The paper proposed a method for video to music generation. There exist not many previous works that utilizes the concept of (video)sequence-to-(music)sequence generation. Most of the previous works tackled the video-to-music generation task as mediaContent-to-sequence task, so the global video features has been used for generating music. Previous works that used the concept of the sequence-to-sequence was mainly studied in the dance-to-music generation task. Therefore, in this paper, the authors tried to generate music using frame-level video information so that each the generated music frame is synchronized with the video frames. To do this, they mainly suggested two techniques which are an visual adaptor module and contrastive video-music pre-training. Visual adaptor aggregates frame-level video features to match with music frames. And, contrastive video-music pre-training is aimed to make synchronization of music beat and video event while preserving overall mood/style coherency. Both the strategies were verified to be effective through the ablation studies.

**Strengths:**

The proposed visual adaptor and pre-training technique with the two negative samplings seems to be working for better modeling synchronization between video and music.

**Weaknesses:**

In Introduction, "Integration of foley and sound effects" paragraph seems not necessary. The paper does not tackles foley sound generation neither sound effects. If we listen to the demo samples, It's more like rhythmic synchronization + bpm modulation, not foley or sound effects.

**Questions:**

In Section 4.1, the authors noted that they used MTG-Jamendo Dataset + 33.7K music tracks from the internet. I think the reason why they have used more 33.7K music tracks from the internet should be explained more in detail.

Also, if we see Appendix, the collected video mostly includes Disney, Tom and Jerry, and Silent Films. I think this fact should also be described in the paper well since the proposed method is valid only on these kind of video contents for now. (The authors mentioned that for the video that contains vocal singing, they excluded the vocal part through source separation technique, however, if there exist some narration or voice of the actors, the proposed technique will not be valid unless they delete speech parts.)

---

> ### Author Response · Authors · 2024-11-21
> **Response to Reviewer JLkN (Part 1/N)**
>
> Thanks for your valuable feedback, and we hope our response fully resolves your concerns.
>
> **[About integration of foley and sound effects]**
>
> > In Introduction, "Integration of foley and sound effects" paragraph seems not necessary. The paper does not tackles foley sound generation neither sound effects. If we listen to the demo samples, It's more like rhythmic synchronization + bpm modulation, not foley or sound effects.
>
> We apologize for any possible misunderstandings caused by this description. The intent of this paragraph was to distinguish the mimicking of foley sounds in the generated music from the logic of traditional foley sound generation. Indeed, our method was not specifically designed for foley sound generation, but many audiovisual artworks employ the technique of mimicking natural sounds with musical instruments to achieve more expressive audiovisual effects. Our method will also possess this capability after training. These foley sound effects, simulated by musical instruments, are actually a special form of music generation, as stated in lines 63-66. Many cases can also be found in the demo samples (for sample, at the beginning of [this sample](https://muvi-v2m.github.io/data/result/video/tom_and_jerry_01[90to110](2).mp4), when Jerry hits Tom in the back of the head with a revolver, this sound is simulated by a set of percussion instruments and a brief string section). Nevertheless, the reviewer JLkN's perspective, which suggests that this does not actually fit the definition of foley sound generation and is essentially just a musical technique, also has merit. Therefore, we will refine this part to emphasize more on musical and instrumental techniques, rather than foley sound generation.
>
> **[About 33.7K music tracks]**
>
> > In Section 4.1, the authors noted that they used MTG-Jamendo Dataset + 33.7K music tracks from the internet. I think the reason why they have used more 33.7K music tracks from the internet should be explained more in detail.
>
> Thank you for pointing this out, it would be clearer if we provide a discussion on our choice. We pre-trained the music generator to first establish a simple yet decent baseline, as we do not want the generator itself to become a bottleneck or critical point for this task. In fact, the model was initially trained using the 33.7K tracks from the internet, but using only these data was not enough to fully meet the baseline standard. It was after this that we noticed the MTG-Jamendo dataset and directly incorporated it into our training set, achieving a reasonably good baseline. Therefore, this was a choice based on experience. To measure the performance of an unconditional generator, we follow previous works [1] and evaluate the methods on the MusicCaps benchmark. Here are preliminary experimental results:
>
> |Methods|FAD|KL|
> |:-:|:-:|:-:|
> |Mousai|7.5|1.59|
> | MusicLM | 4.0 | - |
> | Noise2Music | 2.1 | - |
> | MusicGen (1.5B) | 5.0 | 1.31 |
> | MuVi (uncond. 33.7K) | 8.5 | 2.34 |
> | MuVi (uncond. 50K) | 7.8 | 2.05 |
> | MuVi (uncond. 83.7K) | 7.3 | 1.91 |
>
> We compare the unconditional generator trained with different sets of training data (33.7K from the internet, 50K from the MTG-Jamendo dataset, and the combination), and some baselines (the results are copied from their original papers). For an unconditional generator, it reaches reasonable performance with the combination of the two sets.

---

> ### Author Response · Authors · 2024-11-21
> **Response to Reviewer JLkN (Part 2/N)**
>
> **[About limited video contents]**
>
> > Also, if we see Appendix, the collected video mostly includes Disney, Tom and Jerry, and Silent Films. I think this fact should also be described in the paper well since the proposed method is valid only on these kind of video contents for now.
>
> Thank you for raising this issue. Due to the high production costs of such artistic works, their numbers are very scarce. However, we still managed to find a considerable number of videos, covering a wide range from animation to live-action videos. By combining a video encoder pre-trained on general video data with an adaptor specially designed to create an information bottleneck, we believe this method has a certain degree of general applicability, although this is not the main focus of this paper. We also demonstrated the capability of this method to handle various video content on the demo page.
>
> **[About voice interference]**
>
> > The authors mentioned that for the video that contains vocal singing, they excluded the vocal part through source separation technique, however, if there exist some narration or voice of the actors, the proposed technique will not be valid unless they delete speech parts.
>
> We apologize for the unclearness. In lines 328-329, the "vocals" mentioned in the text include both singing voice and speech. In fact, the tool we used [2] will remove all human voices in the audio.
>
> ---
>
> Once again, thank you for your effort in reviewing our work and your acknowledgment.
>
> **References**
>
> [1] Copet, Jade, et al. Simple and controllable music generation. Advances in Neural Information Processing Systems 36 (2024).
>
> [2] Anjok07 and aufr33. Ultimate vocal remover. https://github.com/Anjok07/ultimatevocalremovergui, 2020.

---

> ### Author Response · Authors · 2024-11-25
> **Looking Forward to Further Feedback**
>
> Dear Reviewer JLkN,
>
> Thank you again for your great efforts and valuable comments.
>
> We have tried to address the main concerns you raised in the review and made huge efforts such as additional experiments. As the end of the rebuttal phase is approaching, we are looking forward to hearing your feedback regarding our answers. We are always happy to have a further discussion and answer more questions (if any).
>
> Thanks in advance,
>
> Submission10758 Authors

---

> > ### Comment · Reviewer_JLkN · 2024-11-26
> > **I will hold my rating score**
> >
> > My main concern still lies on the generalizability of the proposed method (the rest concerns has been resolved). I think the proposed synchronization only works on monophonic like simple instrumental sounds generation now rather than complex multitrack music generation. And, there might be an extensive extra work to apply the proposed method for complex multitrack music generation. So, I think the authors should narrow down the scope of the work to be "certain types of (instrumental) sounds" generation rather than "music" generation.

---

### Meta-Review · Area_Chair_qJ6U · 2024-12-21

**Metareview:**

This paper, MuVi, proposes a new video-to-music generation framework focusing on two primary goals: semantic alignment (music content changing in pair with the video’s mood or scene) and rhythmic synchronization (musical beats matching a video’s pacing).

## Reviewers’ Feedback ##

**Positive Aspects:**

Reviewer cqcQ noted that video-to-music remains a relatively new area, agreed that the discussions on semantic synchronization were sufficient in the rebuttal, and was satisfied with the additional baseline comparison. Reviewer yVkt acknowledged that some of the concerns have been resolved including writing, semantic alignment metric, and additional experimental comparison and benchmarks. Reviewer JLkN noted that most concerns regarding method clarity and experimental comparisons had been addressed.

**Unresolved Issues:**

1. Novelty:  Reviewer yVkt found the authors’ arguments about being the first to tackle local or fine-grained alignment unconvincing, pointing out that there are several relevant methods

2. Generalizability: Reviewer yVkt and Reviewer JLkN noted that, despite extra benchmarks in the rebuttal, the results can not convincingly demonstrate the generalizability of the proposed approach.

3. Metrics & Comparison: There were inconsistencies between MuVi’s definitions of BCS/BHS and those in referenced works. The reviewer yVkt felt this may mislead readers about how results compare with other studies. They also noted that the demo page lacked updated comparative results.

4. Overclaim in In-context learning section: Reviewer cqcQ suggested that the paper’s in-context learning part might not rise to the level of a distinct contribution.

## Recommendation ##

After rebuttal and discussion, the reviewers had mixed opinions (scores of 8, 6, and 3). The reviewer gave it an 8 highlighting the paper’s writing and recognized that video-to-music generation remains a relatively new field, highlighting certain differences from existing methods. I appreciate the efforts of the authors to provide extensive responses and for the reviewers to engage in discussions. However, after rebuttal and discussion, the other two reviewers still had concerns—specifically about overlaps with prior local alignment approaches and insufficient demonstration for broader generalization. Despite one positive review, I recommend rejection due to insufficient novelty and limited applicability.

I encourage the authors to address these concerns and consider resubmitting to a future venue.

**Additional Comments On Reviewer Discussion:**

The rebuttal and discussion phase highlighted several critical concerns: overlap with existing approaches, limited generalization, and potential overclaims. While one reviewer remained positive, highlighting the clear writing and the paper's contribution to a relatively new task, the other two remained unconvinced. They felt the core ideas overlapped with prior frameworks (Reviewer yVkt) and that generalizability was not adequately demonstrated (Reviewers yVkt and JLkN). Despite the authors' responses, including additional comparisons and clarifications, these core concerns are still there. Although video-to-music generation is an interesting and challenging task, I concur with the reviewers' concerns and recommend rejection.

---

### Decision · Program_Chairs · 2025-01-22

Reject